# Causal Climate Emulation with Bayesian Filtering

**Sebastian Hickman**[1,4,*]    **Ilija Trajkovic**[2]    **Julia Kaltenborn**[3,4]    **Francis Pelletier**[4]

**Alex Archibald**[1]    **Yaniv Gurwicz**[5]    **Peer Nowack**[2]    **David Rolnick**[3,4]    **Julien Boussard**[3,4]

[1]Yusuf Hamied Department of Chemistry, University of Cambridge, UK
[2]Institute of Theoretical Informatics, Karlsruhe Institute of Technology, Germany
[3]School of Computer Science, McGill University, Canada
[4]Mila - Quebec AI Institute, Canada
[5]Intel Labs, Israel
[*]now at the European Centre for Medium-Range Weather Forecasts, UK

## Abstract

Traditional models of climate change use complex systems of coupled equations to simulate physical processes across the Earth system. These simulations are highly computationally expensive, limiting our predictions of climate change and analyses of its causes and effects. Machine learning has the potential to quickly emulate data from climate models, but current approaches are not able to incorporate physically-based causal relationships. Here, we develop an interpretable climate model emulator based on causal representation learning. We derive a novel approach including a Bayesian filter for stable long-term autoregressive emulation. We demonstrate that our emulator learns accurate climate dynamics, and we show the importance of each one of its components on a realistic synthetic dataset and data from two widely deployed climate models.

## 1   Introduction

In order to respond to climate change, it is necessary to understand future climates as precisely as possible, under different scenarios of anthropogenic greenhouse gas emissions. Earth system models (ESMs), which couple together complex numerical models simulating different components of the Earth system, are the primary tool for making these projections, but require solving vast numbers of equations representing physical processes and are thus computationally expensive [Balaji et al., 2017, Mansfield et al., 2020]. This severely limits the number of future scenarios that can be simulated and attribution studies investigating the influence of climate change on extreme events. Reduced physical models, such as Simple Climate Models [Leach et al., 2021], are commonly used to speed up the process, but at the expense of oversimplifying necessary complex processes.

In contrast, data-driven models are flexible and efficient but often lack the physical grounding of Simple Climate Models [Watson-Parris et al., 2022], as they rely on correlations in data. Deep learning (DL) models have shown considerable success in medium-range weather forecasting up to a few weeks ahead [Bi et al., 2023, Lam et al., 2023, Lang et al., 2024, Bodnar et al., 2024], but many such models become physically unrealistic or unstable when performing long-term climate projections [Chattopadhyay et al., 2024, Karlbauer et al., 2024]. Recent approaches for climate emulation [Watt-Meyer et al., 2024, Cachay et al., 2024, Guan et al., 2024, Cresswell-Clay et al., 2024] have achieved stable long-term simulations suitable for climate modeling, with small biases in climate statistics. However, it is unclear whether these models learn physical climate processes, and they may not always respect known causal relationships [Clark et al., 2024].

A potential reconciliation of the respective advantages of physics-based and data-driven models lies in causal discovery, the automatic discovery of causal dependencies from available data [Runge et al., 2023, Rohekar et al., 2021], which has emerged as a valuable tool to improve our understanding of physical systems across various fields, including Earth sciences [Runge et al., 2019a,b]. Many

causal representation learning tools have been developed recently, especially after Zheng et al. [2018] introduced a fully differentiable framework to learn causal graphs using continuous optimization. This framework was extended to derive Causal Discovery with Single-parent Decoding (CDSD), a causal representation learning method for time series [Brouillard et al., 2024].

In this work, we move beyond CDSD to develop a physically-guided model for causal climate emulation and demonstrate that it effectively captures important climate variability, generates stable long-term climate projections, and provides the capacity for counterfactual experiments. This work is a step towards trustworthy, physically-consistent emulation of climate model dynamics.

**Main contributions**:

- We develop a novel machine learning algorithm for causal emulation of climate model dynamics, incorporating (i) additional losses that ensure invariant properties of the data are preserved, (ii) a Bayesian filter that allows for stable long-term emulation when autoregressively rolling out predictions.
- We demonstrate the performance of our model on a synthetic dataset that mimics atmospheric dynamics and on a dataset from a widely deployed climate model, and we perform ablation studies to show the importance of each component of the model.
- We perform counterfactual experiments using our causal emulator, illustrating its capacity to model the physical drivers of climate phenomena, and its interpretability.

## 2   Related work

### 2.1   Causal learning for time series

Causal discovery for time series data can be approached as a constraint- or score-based task. Constraint-based methods [Runge, 2015, Rohekar et al., 2023] build a graph by iteratively testing all known variable pairs for conditional independence, and have been widely used in climate science [Debeire et al., 2025], particularly PCMCI [Runge, 2018, Runge et al., 2019a, Runge, 2020, Gerhardus and Runge, 2020]. PCMCI can be combined with dimensionality reduction methods to obtain low-dimensional latent variables before learning causal connections between these latents [Nowack et al., 2020, Tibau et al., 2022, Falasca et al., 2024]. Constraint-based methods scale poorly and become intractable for high-dimensional data, especially when considering non-linear relationships. Score-based methods address this issue by finding a graph that maximizes the likelihood of observed data, using differentiable acyclicity constraints to ensure identifiability [Zheng et al., 2018]. These methods have been extended to time series data, and non-linear data [Pamfil et al., 2020, Sun et al., 2023]. However, they only account for causal connections between observed variables and do not tackle the problem of learning a causal latent representation from high-dimensional data.

Causal representation learning aims to address this gap. Under certain conditions, it is possible to disentangle relevant latent variables from high-dimensional observed data [Hyvarinen and Morioka, 2017, Hyvarinen et al., 2019, Khemakhem et al., 2020]. Recent work demonstrated that a latent representation and a causal graph between latent variables can be learned simultaneously [Lachapelle et al., 2020, Schölkopf et al., 2021] via a fully differentiable framework [Zheng et al., 2018, Brouillard et al., 2020]. CDSD [Brouillard et al., 2024, Boussard et al., 2023] aims to recover both the latent variables and the temporal causal graph over these latents from a high-dimensional time series, where the causal graph is shown to be identifiable under the strong *single-parent decoding assumption* – i.e. each observed variable is mapped to a single latent, while each latent can be the parent of multiple observed variables.

In our setting, this assumption imposes that the latents correspond to single climate modes in a geographic region which interact through the causal graph. While strong, the assumption is well-suited to modeling long-range climate dynamics and teleconnections, where variations in distinct regions around the world can causally affect the climate in other regions. The learned latents mirror simplified climate indices (e.g., scalar indices representing climate variables in fixed regions) that climate scientists widely rely on to describe large-scale atmospheric dynamics. Climate scientists often use simple representations of complex processes for interpretable, parsimonious description of large-scale phenomena. Our work parallels such methods closely. Furthermore, the single-parent assumption is a principled first step towards interpretable and trustworthy data-driven climate models. It makes our tool interpretable for climate scientists and gives theoretical guarantees of

causal identifiability. This allows exploration of the effect of interventions through counterfactual experiments, which we hope will be useful for attribution studies by domain scientists.

## 2.2 Climate model emulation

Recent years have seen extensive work on developing large deep learning models for Earth sciences. Focus has been placed on medium-range weather forecasting [Keisler, 2022, Bi et al., 2023, Lam et al., 2023, Lang et al., 2024, Bodnar et al., 2024, Pathak et al., 2022], but a growing body of work has considered long-term climate projections (a much longer time horizon than weather), where physical realism and out-of-distribution generalization to unseen future scenarios are primary challenges.

Building on the success of deep learning-based weather forecasts, several studies have considered autoregressively rolling out shorter-term prediction models, typically trained to predict the next timestep at 6 or 12 hour intervals, to emulate climate models on longer time-scales [Nguyen et al., 2023, Duncan et al., 2024, Cresswell-Clay et al., 2024]. Most are deterministic models based on U-Net or SFNO architectures [Ronneberger et al., 2015, Bonev et al., 2023, Wang et al., 2024], with one recent study employing a diffusion model to make probabilistic climate projections [Cachay et al., 2024]. In all these cases, generating stable long-term projections is difficult, as the models do not necessarily capture physical principles governing long-term climate dynamics and often require careful parameter tuning and design choices [Guan et al., 2024]. Watt-Meyer et al. [2024] incorporate simple corrections to preserve some physical consistency, while Guan et al. [2024] focus on matching a physical quantity, the *spatial spectrum*, to improve stability of long-term rollouts.

## 2.3 Modeling spatiotemporal climate dynamics

More principled statistical approaches aim at learning a dynamical system from the data [Yu and Wang, 2024]. These approaches integrate physical principles as inductive biases or constraints, aiming to automatically discover relationships between different variables [Shen et al., 2024]. Similar to our work, these approaches focus on modeling the underlying climate dynamics to capture natural oscillations and stochasticity in the climate system (internal climate variability) and inform seasonal-to-decadal predictions [Van den Dool et al., 2006, Rader and Barnes, 2023, Cosford et al., 2025]. These approaches have often focused on specific known climate phenomena [Spuler et al., 2025] and have been used for attribution studies to determine the causal drivers of these phenomena [Sippel et al., 2024]. Here, we use causal representation learning to model internal climate dynamics, predict natural variability in temperature around the world, and enable attribution studies.

## 3 Methods

To meet the challenges of learning causal structure, physical realism, and interpretability, we introduce the PICABU model (Physics-Informed Causal Approach with Bayesian Uncertainty). We describe each aspect of this approach here, and the overall pipeline is illustrated in Figure 1, showing the latent representation, the loss used during training, the Bayesian filter, and counterfactual experiments.

### 3.1 Core framework

Our framework leverages CDSD [Brouillard et al., 2024], which jointly learns latent variables $\mathbf{Z}$ and a mapping from $\mathbf{Z}$ to the high-dimensional observations $\mathbf{X}$, and proves identifiability of the causal graph under the *single-parent assumption* (for details, see Appendix B). This latent-to-observation graph is parameterized by a matrix $W$ and neural networks. Latents at time $t$ are mapped to latents at previous timesteps through a directed acyclic graph (DAG) and neural networks. The parameters are learned by maximizing an evidence lower bound (ELBO). The augmented Lagrangian method (ALM, Nocedal and Wright, 2006) is used to optimize the loss under the constraint $\mathcal{C}_{\text{single-parent}}^{\{\lambda,\mu\}}$ and ensure that $W$ is orthogonal and satisfies the assumptions needed for identifiability. A penalty $\mathcal{P}_{\text{sparsity}}^{\lambda}$ is added to the loss to sparsify the DAG and retain only causally-relevant links between latents.

Unlike in CDSD, we constrain ($\mathcal{C}_{\text{sparsity}}^{\{\lambda,\mu\}}$) rather than penalize ($\mathcal{P}_{\text{sparsity}}^{\lambda}$) the number of causal connections, to encode domain knowledge in the causal graph [Trenberth et al., 1998] and improve model convergence [Gallego-Posada et al., 2022]. Below, the terms incorporate the corresponding ALM

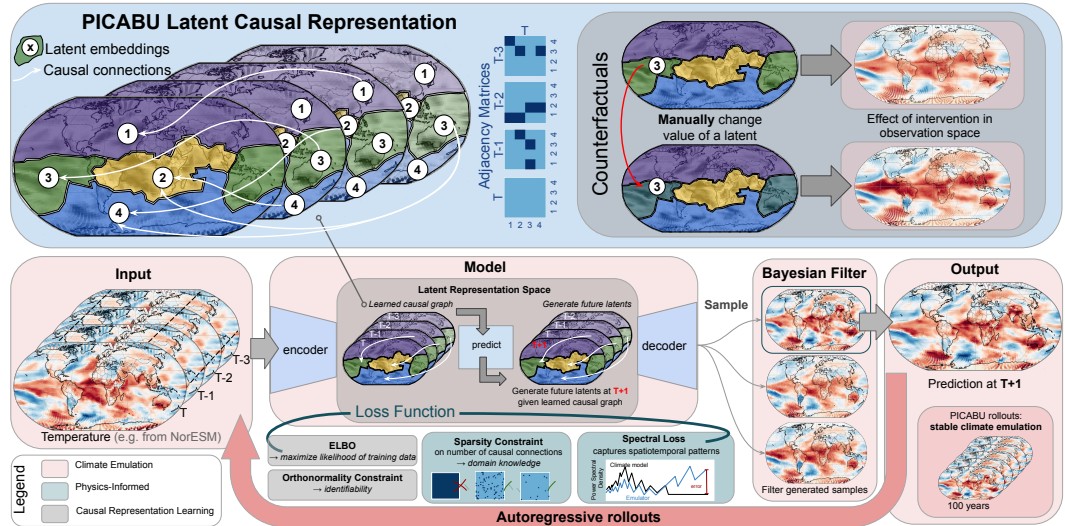

Figure 1: **High-level schematic of the PICABU pipeline.** The key features of the PICABU pipeline are illustrated: the latent embeddings under the single-parent assumption, the learned causal graph over these latents, the loss used during training, the Bayesian filter allowing for stable autoregressive rollouts, and the possibility of counterfactual experiments.

coefficients, $\{\lambda, \mu\}$.

$$\mathcal{L}_{\text{core}}^{\{\lambda,\mu\}} = \text{ELBO} + \mathcal{C}_{\text{single parent}}^{\{\lambda,\mu\}} + \mathcal{C}_{\text{sparsity}}^{\{\lambda,\mu\}}, \tag{1}$$

### 3.2 Additional loss functions

To improve the predictions of the learned model for longer rollouts, we include a second loss term, the continuous ranked probability score (CRPS; Matheson and Winkler [1976], Hersbach [2000]), a measure of probabilistic next-timestep prediction accuracy.

$$\mathcal{L}_{\text{CRPS}} = \int_{-\infty}^{\infty} \left[ F(\hat{\mathbf{X}}) - \mathbf{1}_{\{\mathbf{X} \leq \hat{\mathbf{X}}\}} \right]^2 d\hat{\mathbf{X}} \tag{2}$$

$\mathbf{X}$ is the true next timestep from the training data and $F(\hat{\mathbf{X}})$ is the cumulative distribution function of the predicted next timestep. Furthermore, to ensure that the model preserves the spatiotemporal structure of the data, we incorporate loss terms measuring the $\ell^1$ difference between the true and inferred spatial spectrum of the data, and between the true and inferred temporal spectra for each grid cell:

$$\mathcal{L}_{\text{spatial}} = \sum_k |\mathcal{F}_{spat}[\mathbf{X}](k) - \mathcal{F}_{spat}[\hat{\mathbf{X}}](k)|; \quad \mathcal{L}_{\text{temporal}} = \sum_j |\mathcal{F}_{temp}[\mathbf{X}](j) - \mathcal{F}_{temp}[\hat{\mathbf{X}}](j)| \tag{3}$$

where $\mathcal{F}_{spat}$ denotes the spatial Fourier transform, $k$ the spatial frequencies, $\mathcal{F}_{temp}$ the temporal Fourier transform, and $j$ the temporal frequencies. We choose these quantities as learning accurate power spectra is necessary for realistic emulation. Learning higher frequencies for decaying spectra is difficult with neural networks [Rahaman et al., 2019], and the $\ell^2$ norm suffers from small gradients when learning wavenumbers associated with high frequencies [Chattopadhyay et al., 2024]. We therefore use the $\ell^1$ norm across the whole spectrum [Shokar et al., 2024].

The full training objective is as follows, with $\lambda_{\text{CRPS}}$, $\lambda_{\text{s}}$ and $\lambda_{\text{t}}$ the penalty coefficients:

$$\mathcal{L}_{\text{core}}^{\{\lambda,\mu\}} + \lambda_{\text{CRPS}}\mathcal{L}_{\text{CRPS}} + \lambda_{\text{s}}\mathcal{L}_{\text{spatial}} + \lambda_{\text{t}}\mathcal{L}_{\text{temporal}} \tag{4}$$

We find that the coefficients of the penalty terms and the initial values of the ALM for the constraints need to be carefully chosen to ensure effective model training. Appendix O discusses the coefficients and hyperparameter choices, and gives the values used in our experiments.

## 3.3 Bayesian filtering for stable autoregressive rollouts

Once trained, our model gives a distribution over latent variables given the latents at previous timesteps: $p(\mathbf{z}^t|\mathbf{z}^{<t})$ (the learned transition model). To perform long-term climate prediction, we use an autoregressive model that predicts the next timestep sequentially. Rather than predicting at each step $\mathbf{z}^t$ from the mean of $p(\mathbf{z}^{<t})$, we perform recursive Bayesian estimation and sample $N$ values from $p(\mathbf{z}^t|\mathbf{z}^{<t})$, to preserve the full distribution through time. Furthermore, as each latent is mapped to a prediction in the observation space, we add a particle filtering step. For each sample $\mathbf{z}_{\tilde{n}}^{\leq t}$ at time $\leq t$, we sample $R$ values from $p(\mathbf{z}^{t+1}|\mathbf{z}_{\tilde{n}}^{\leq t})$, and assign a likelihood to each of them, which encode whether the samples are mapped to an observation satisfying the desired statistics of the data. By keeping the sample with the highest likelihood value, we prevent error accumulation or convergence to the mean and obtain a set of $N$ stable trajectories from our autoregressive rollout.

When performing recursive Bayesian estimation for a state-space model, we typically aim to compute the posterior distribution for the latents given a sequence of observations as follows:

$$p(\mathbf{z}^t|\mathbf{x}^{\leq t}) \propto p(\mathbf{x}^t|\mathbf{z}^t)p(\mathbf{z}^t|\mathbf{x}^{\leq t-1}). \tag{5}$$

To sample from this posterior distribution, we can estimate $p(\mathbf{z}^t|\mathbf{x}^{<t})$ (learned encoder model). However, in our context we do not observe $\mathbf{x}^t$ as we are performing prediction and cannot sample from $p(\mathbf{x}^t|\mathbf{z}^t)$. We therefore propose using the Fourier spatial spectrum at each timestep as a proxy for our observations: we assume that it is constant through time and treat its value as the true observation. We then compute $p(\tilde{\mathbf{x}}|\mathbf{z}^t)$ where $\tilde{\mathbf{x}}$ is the constant spatial spectrum of the climate variable over Earth, obtained from available data. We then sample from the posterior distribution $p(\mathbf{z}^t|\tilde{\mathbf{x}}^{\mathbf{t}}, \mathbf{z}^{<t})$, and eliminate samples with low importance score: if a sample corresponds to a spatial spectrum that is very far from the true spatial spectrum, it will be assigned a low probability and thus will be discarded. We represent $p(\tilde{\mathbf{x}}|\mathbf{z}^t)$ using a Laplace distribution $\mathcal{L}(\tilde{\mathbf{x}}^t, \tilde{\sigma})$. The mean $\tilde{\mathbf{x}}^t$ is obtained by decoding $\mathbf{z}^t$ and mapping them to observations $\mathbf{x}^t$, which are then Fourier-transformed to obtain $\tilde{\mathbf{x}}^t$. The variance $\tilde{\sigma}$ is estimated directly from the observations and assumed constant through time. Contrary to the standard Bayesian filter approach, we choose not to use the variance estimated by the model, which is empirically higher than the observations' variance. The choice of constant variance helps to constrain the model and obtain better projections. More details on this choice can be found in Appendix J.

In practice, one could use any known statistic of the data. We focus on the spatial spectrum as predicting an accurate spatial spectrum is necessary to prevent models from propagating errors during rollouts [Chattopadhyay et al., 2024, Guan et al., 2024]. This approach could be applied to many statistics known to be constant or invariant through time, including conserved quantities [White et al., 2024]. Once the spatial spectrum, assumed fixed, is computed by taking the mean of spatial spectra across observed NorESM2 data, our autoregressive rollout proceeds as given in Algorithm 1.

---

**Algorithm 1** Autoregressive rollout with Bayesian filtering

---

**Input:** Observations $\mathbf{x}^{\leq T}$, trained encoder $p(\mathbf{z}^{\leq t}|\mathbf{x}^{\leq t})$, trained decoder $\mathbf{x} = f(\mathbf{z})$, learned transition model $p(\mathbf{z}^t|\mathbf{z}^{<t})$, ground truth spatial spectrum $\tilde{\mathbf{x}}$ with standard deviation $\tilde{\sigma}$, number of sampled trajectories $N$ and sample size $R$, prediction time range $m$
**Initialization:** Get $N$ samples $\mathbf{z}_{\tilde{n}}^{\leq T} \sim p(\mathbf{z}^{\leq T}|\mathbf{x}^{\leq T})$
**for** $i = 1$ **to** $m$ **do**
  Sample, for each $n \leq N$, $R$ samples $\mathbf{z}_{n,r}^{T+i} \sim p(\mathbf{z}^{T+i}|\mathbf{z}^{<T+i})$
  For each $n \leq N, r \leq R$, decode $\mathbf{x}_{n,r}^{T+i} = f(\mathbf{z}_{n,r}^{T+i})$ and use FFT to get the associated spectrum $\tilde{x}_{n,r}^{T+i}$ and weights $w_{n,r}^{T+i} = \mathcal{L}(\tilde{\mathbf{x}}|\tilde{\mathbf{x}}_{n,r}^{T+i}, \tilde{\sigma})$
  For each $n \leq N$, sample one value from $\{\mathbf{z}_{n,r}^{T+i}\}$ using unnormalized weights $\{w_{n,r}^{T+i} \cdot p(\mathbf{z}_{n,r}^{T+i}|\mathbf{z}_n^{<T+i})\}$
**end for**
**Output:** $N$ latent trajectories $\{\mathbf{z}_{n \leq N}^{T \leq t \leq T+m}\}$

---

## 4 Experiments

### 4.1 Recovering a known causal graph

The spatially averaged vector autoregressive (SAVAR) model [Tibau et al., 2022] allows the creation of time series with known underlying causal relations, designed to benchmark causal discovery methods. The model generates data following an autoregressive process, with modes of variability interacting through defined connections mimicking those in the climate system [Nowack et al., 2020]. It exhibits two essential properties of climate models: spatial aggregation (each grid cell is influenced by neighboring cells) and both local and long-range dependencies. Although it is a simplified model that does not fully capture real-world climate systems, it provides a controlled setting with a known ground truth causal structure. Thus, SAVAR allows us to rigorously assess the ability of PICABU to accurately recover causal relationships from high-dimensional data.

**Data.** We generate several datasets with $N = 4, 25,$ or $100$ latents, and four varying levels of difficulty. In the "easy" dataset, latents depend only on themselves at one previous timestep. In the "med-easy" dataset, they additionally depend on each latent at a previous timestep with probability $p = 1/(N-1)$. This probability grows to $p = 2/(N-1)$ for the "med-hard" and $p = 1/2$ for the "hard" dataset. The causal dependencies are restricted to the five previous timesteps. The expected number of edges in the true graph is thus $M = N + N \cdot (N-1) \cdot p$. The strength values of the autoregressive causal links are drawn from a beta distribution $\mathcal{B}(4, 8)$ with mean $1/3$ and standard deviation $0.13$, chosen to keep most link strengths moderate and away from 0 or 1, and are then normalized. The generative process of the SAVAR data is linear. Noise is added to the datasets with variance equal to the signal variance. SAVAR datasets are illustrated in Appendix C.

**Models.** We compare PICABU to two baseline methods: CDSD [Brouillard et al., 2024] and Varimax-PCMCI (V-PCMCI) [Runge et al., 2015]. We report in Table 1 the F1-score $\frac{2}{\text{recall}^{-1}+\text{precision}^{-1}}$ for all three methods. The F1-score ranges from 0 to 1, indicating how well the true graph is learned, with 1 indicating a perfect graph. For all methods, we set the number of latent variables equal to the true latent dimension of the SAVAR datasets and use linear transition and decoding models. We run PICABU without the spectral components of the loss (i.e. $\lambda_s = \lambda_t = 0$) as the data is not complex enough to require additional penalties. For a fair comparison, all models retain the true number of edges $M$ in their graphs. As V-PCMCI scales poorly, we do not report results on the 100 latent datasets due to the extensive computational resources this experiment would require. The details of the training procedure and hyperparameter search are given in Appendix C.2.

| N latents | 4 | | | | 25 | | | | 100 | | | |
|---|---|---|---|---|---|---|---|---|---|---|---|---|
| Difficulty | Easy | Med easy | Med hard | Hard | Easy | Med easy | Med hard | Hard | Easy | Med easy | Med hard | Hard |
| PICABU | **1** | **1** | **1** | **1** | **1** | 0.97 | 0.95 | 0.81 | **0.98** | **0.99** | 0.95 | **0.84** |
| CDSD | **1** | **1** | **1** | 0.82 | 0.96 | **1** | **0.96** | **0.83** | **0.98** | 0.98 | **0.96** | 0.80 |
| V-PCMCI | **1** | 0.8 | 0.48 | 0.66 | 0.76 | 0.44 | 0.42 | 0.22 | - | - | - | - |

Table 1: **PICABU recovers the true SAVAR causal graph with high accuracy.** F1 score of three methods, PICABU, CDSD, and V-PCMCI on 12 SAVAR datasets with three different dimensionalities (N latents $4, 25, 100$) and four difficulty levels (easy, med-easy, med-hard, hard) corresponding to the increasing number of expected edges in the graph.

**Results.** PICABU achieves high F1-scores on all datasets of various dimensionalities and difficulties, and performs slightly better than CDSD. Both methods disentangle the latents very well in all datasets. The additional losses are not needed here to achieve high accuracy, as the SAVAR datasets are of relatively low complexity. When tested on climate model data, CDSD predictions diverge (see §4.2), showing the fundamental importance of the additional loss terms when modeling real climate model data. Another benefit of PICABU is the possibility to control the final sparsity of the graph directly via a constraint. CDSD, in contrast, requires careful early-stopping or a thorough hyperparameter search over its sparsity penalty coefficient.

V-PCMCI achieves lower accuracy, even on datasets of lower complexity or latent dimensionality. We argue that because the latents are learned using Varimax-PCA before the causal connections are learned, it might not disentangle the latents correctly, especially as noise increases (Appendix C.3).

## 4.2 Pre-industrial climate model emulation

**Data.** To assess performance in emulating widely used climate models, we evaluate PICABU on monthly NorESM2 [Seland et al., 2020] and CESM2-FV2 [Danabasoglu et al., 2020] surface temperature data at a resolution of 250 km, re-gridded to an icosahedral grid. In this section, we focus on pre-industrial data, using 800 years of data provided by two NorESM2 simulations initialized with different initial conditions. We normalize and deseasonalize the data by subtracting the monthly mean and dividing by the monthly standard deviation at each grid point. Figure 10 shows an example of six successive timesteps. The 800 years of data are split into 90% train and 10% test sets. Results for NorESM2 are reported below; for CESM2-FV2, refer to Appendix I.

**Models.** PICABU is trained to take five consecutive months as input and predict the next month. An example prediction is shown in Figure 2. V-PCMCI learns a causal graph and a mapping from latents to observables, but not the full transition model. We enhance it by fitting, for each latent, a regression on the previous latents it causally depends on, according to the learned PCMCI graph. This allows us to predict the next timestep, given the five previous timesteps. We also train a Vision Transformer [Dosovitskiy et al., 2021] with a positional encoding adapted from Nguyen et al. [2023], referred to as "ViT + pos. encoding". Details on this model are given in Appendix D. Furthermore, we perform a study in which we individually ablate each term in PICABU's loss. After training, we sample initial conditions from the test set and, for each initial condition, autoregressively project 100 years forward. PICABU, ablated models, and V-PCMCI use our Bayesian filter, whereas "ViT + pos. encoding" does not, as it is deterministic.

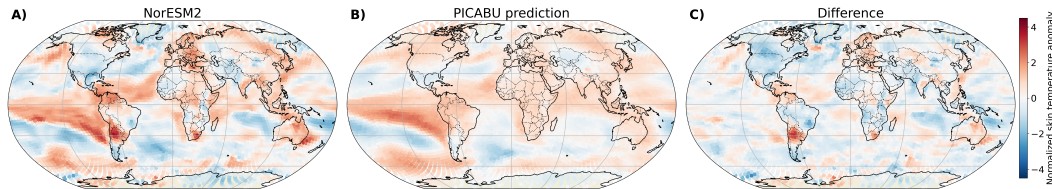

Figure 2: **An example next timestep PICABU prediction.** A) normalized temperature of NorESM2 (target), B) prediction from PICABU for the target month, C) difference between target and prediction. All data is on an icosahedral grid, normalized, and deseasonalized.

**Results.** As climate model data is chaotic, high-dimensional, and driven by complex spatiotemporal dynamics, we seek a model that has both low mean bias over long rollouts and captures the spatiotemporal variability of the data. To capture these properties, we evaluate the mean, standard deviation, range, and log-spectral distance (LSD) of key climate variables, with the goal being for ML models to come as close as possible to the ground truth values for each of these statistics. We compute these values for two climate variables of great relevance to long-term climate dynamics: the normalized global mean surface temperature (GMST) and the Niño3.4 index, which represents the El Niño Southern Oscillation (ENSO) (the most important natural source of variability in the climate system). The GMST and Niño3.4 index statistics are computed over ten 50-year periods from NorESM2 pre-industrial data. In Table 2, we compare values for the ground truth NorESM2 data and the models PICABU, V-PCMCI, "ViT + pos. encoding", and ablations.

Overall, we find that PICABU comes closest to matching the ground truth across all statistics for the GMST and Niño3.4 indices. By contrast, CDSD diverges and is unable to fit the data at all. V-PCMCI gives predictions with unrealistically low spatiotemporal variability (as captured in the std. dev. and range being far from ground truth values) for both climate variables, and the ViT with positional encoding exhibits overly high variability, especially for GMST. Our tests also affirm the value of the different PICABU components. Removing the CRPS term leads to a higher range in GMST, while removing the spectral term leads to much higher bias. Removing the orthogonality constraint leads to poorer performance for GMST and comes at the expense of identifiability and interpretability. PICABU without the Bayesian filter or sparsity term both diverge.

| | GMST | | | | Niño3.4 | | | |
|---|---|---|---|---|---|---|---|---|
| | **Mean** | **Std. Dev.** | **Range** | **LSD** | **Mean** | **Std. Dev.** | **Range** | **LSD** |
| Ground truth | 0 | 0.177 | 1.357 | 0 | 0 | 0.927 | 5.73 | 0 |
| CDSD | Diverges | - | - | - | Diverges | - | - | - |
| V-PCMCI | 0.0676 | 0.078 | 0.73 | 0.250 | 0.196 | 0.346 | 3.45 | 0.180 |
| ViT + pos. encoding | -0.0036 | 0.458 | 4.91 | 0.442 | -0.0053 | 0.620 | 6.26 | 0.326 |
| Unfiltered | Diverges | - | - | - | Diverges | - | - | - |
| No sparsity | Diverges | - | - | - | Diverges | - | - | - |
| No ortho | 0.0624 | 0.527 | 3.28 | 0.583 | 0.0867 | 0.987 | 5.94 | 0.267 |
| No spectral | 0.0796 | 0.619 | 3.81 | 0.658 | -0.225 | 1.52 | 9.34 | 0.334 |
| No CRPS | -0.0424 | 0.536 | 3.65 | 0.595 | 0.0148 | 0.953 | 5.80 | 0.300 |
| PICABU | -0.0518 | 0.277 | 2.02 | 0.444 | -0.0914 | 1.05 | 5.99 | 0.191 |

Table 2: **PICABU shows a small bias and captures climate variability better than comparable methods.** The left section shows the mean, standard deviation, range, and log spectral distance (LSD) of the GMST, and the same for the Niño3.4 index on the right. The first row shows the values of the metrics for the ground truth NorESM2, followed by comparison methods, and then PICABU ablations. For all metrics, being closer to the ground truth is better.

We also consider the power spectral density (PSD, Froyland et al. [2021]) of the Niño3.4 index and GMST for NorESM2 data and the different models (Figure 3). We find that PICABU captures the temporal dynamics of Niño3.4 index and GMST better than all other models. In particular, the power spectrum of the Niño3.4 index shows a peak at a period of three years, and the power spectrum of PICABU closely approximates the ground truth around this peak, illustrating that the model has learned accurate climate dynamics, while the other models do not perform as well (Figure 11).

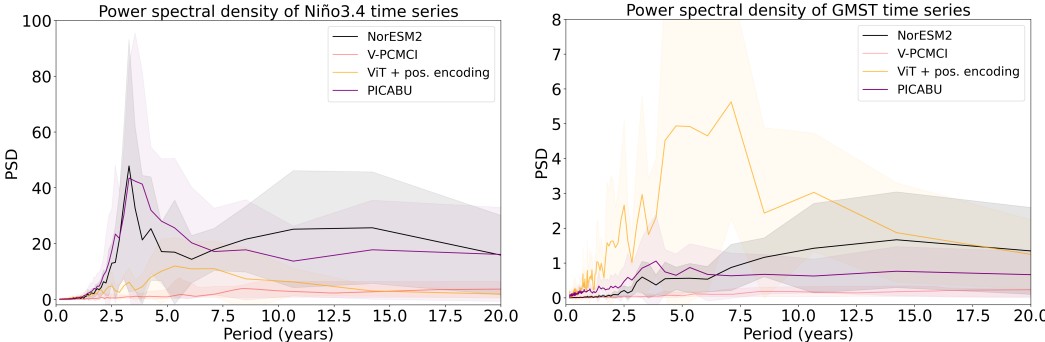

Figure 3: **PICABU learns accurate temporal variability for ENSO, and outperforms other methods in learning GMST variability.** We run PICABU for ten different 50-year emulations, and compute the mean and standard deviation of the spectra, doing the same for ten different 50-year periods of NorESM2 data. On the left, the power spectra for the Niño3.4 index, for PICABU, the ViT, V-PCMCI, and ground truth data (NorESM2) are shown. The same is shown for GMST on the right.

However, we note that, while outperforming the other models, PICABU exhibits too much power at very high frequencies for Niño3.4 and GMST (Appendix F, Figure 12), suggesting room for further improvement in emulating large-scale variability at short timescales. Similar plots for the ablated PICABU models are given in Figure 4, with all ablations showing a decrease in performance, illustrating the importance of the additional loss terms.

Results for two further important known patterns of variability, the Indian Ocean Dipole (IOD) and the Atlantic Multidecadal Oscillation (AMO) [Saji et al., 1999, Kerr, 2000, McCarthy et al., 2015], are shown in Appendix H, and are consistent with the above results for ENSO and GMST. Additionally, performance on the CESM2-FV2 climate model (Appendix I) is similar to the results for NorESM2

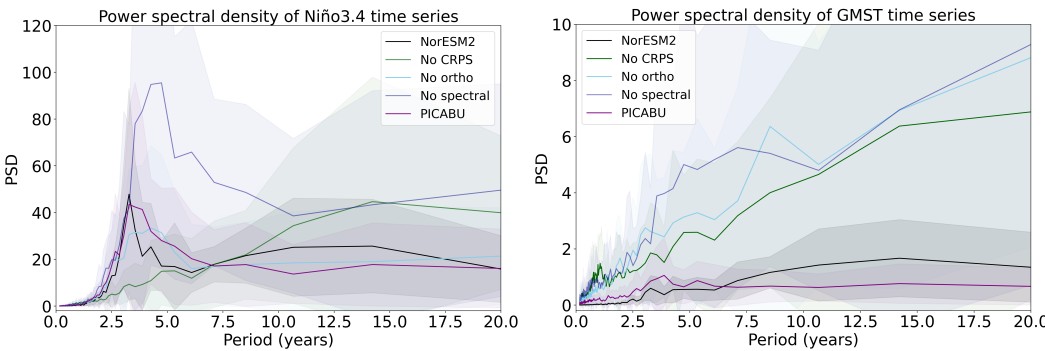

Figure 4: **PICABU outperforms ablated models in learning ENSO and GMST variability.** We run PICABU for ten different 50-year emulations, and compute the mean and standard deviation of the spectra, doing the same for ten different 50-year periods of NorESM2 data. On the left, the power spectra for the Niño3.4 index, for PICABU and ablations, and ground truth data (NorESM2) are shown. On the right, the same is shown for GMST.

data. To complement Table 2, the average annual temperature range for each grid cell is calculated for both NorESM2 and for PICABU, and compared in Figure 14. The spatial pattern of the annual range of projected temperatures is reasonable, but has a greater range in some regions (Appendix G).

## 4.3 Learning a causal model improves PICABU's generalization performance

On pre-industrial data, we train PICABU and two ablated models, where the sparsity and orthogonality constraints are separately dropped from the loss (Appendix M). We evaluate these models by examining their next-timestep predictions of surface temperature when provided with 256 random initial conditions from three different emissions scenarios, sampled from SSP2-4.5, SSP3-7.0, and SSP5-8.5 between years 2070 and 2100. These correspond to a considerably warmer climate than the pre-industrial data used for training. Note that we do not include the anthropogenic emissions as inputs to the emulator; instead, these experiments evaluate the ability of the models to generalize to a new distribution of initial conditions. Table 3 shows the mean absolute error (MAE), coefficient of determination ($R^2$), and LSD for next timestep predictions.

| Scenario | picontrol | | | SSP2-4.5 | | | SSP3-7.0 | | | SSP5-8.5 | | |
|---|---|---|---|---|---|---|---|---|---|---|---|---|
| **Metric** | MAE | $R^2$ | LSD | MAE | $R^2$ | LSD | MAE | $R^2$ | LSD | MAE | $R^2$ | LSD |
| PICABU | **0.70** | **0.11** | 0.059 | **0.72** | **-0.17** | 0.052 | **0.75** | **-0.41** | **0.048** | **0.75** | **-0.91** | **0.031** |
| No sparsity | 0.74* | 0.03* | **0.046*** | 0.77* | -0.30* | **0.042*** | 0.78* | -0.50* | 0.048 | 0.81* | -1.2* | 0.034* |
| No ortho | 0.77* | 0.01* | 0.054* | 0.80* | -0.42* | 0.047* | 0.80* | -0.63* | 0.049 | 0.84* | -1.4* | 0.034* |

Table 3: **Learning a causal graph helps generalization to new emission scenarios.** Next-timestep prediction results when given initial conditions from out-of-distribution data. For each scenario, three metrics are shown: MAE (lower is better), $R^2$ (higher is better), and LSD (lower is better). We evaluate PICABU, PICABU without the sparsity constraint, and PICABU without the orthogonality constraint with initial conditions corresponding to scenarios SSP2-4.5, SSP3-7.0, and SSP5-8.5 (from left to right, diverging further from the training distribution). * indicates a statistically significant difference with the corresponding PICABU score, obtained using a t-test. For $R^2$, the relative difference is taken with respect to 1, reflecting how much further the metric is from the ideal $R^2 = 1$.

For the pre-industrial control data, the models perform relatively similarly, but the full causal model, with both sparsity and orthogonality enforced, performs better for out-of-distribution initial conditions. This suggests that learning the causal dynamics aids out-of-distribution generalization.

## 4.4 Counterfactual experiments for attribution of extreme events

Climate models are crucial for attribution studies. When large anomalies occur, e.g. a particularly hot year, climate scientists typically perturb the inputs to ESMs and observe any changes in the projected climate to attribute causal drivers to these events [Kosaka and Xie, 2013]. This is compute- and time-consuming, particularly if many experiments are needed to capture the probabilistic nature of the simulations. With PICABU, due to its causal interpretability, we can directly intervene on the input data to carry out counterfactual experiments. As a case study, we consider the anomalously warm series of months around November 1851 in the NorESM2 data. We identify the Niño 3.4 index grid cells that correspond to sea surface temperatures in the equatorial eastern Pacific, a widely used indicator of ENSO. We then carry out a counterfactual experiment: we intervene on these grid cells, manually setting their values to new, perturbed values, and observe the resulting change in the next-timestep prediction for November 1851. No other alteration to the observational space, latent variables or causal graph is made.

These intervened grid cells influence the value of their parent latent variables, which then interact through the learned causal graph to generate latents at the next timestep, which are in turn decoded to produce temperature fields. Figure 5 shows that if El Niño had been stronger, then temperatures would have been even higher globally, according to the learned causal model. In Appendix P, we illustrate additional interventions on Alaskan and IOD temperatures. We also intervene in latent space rather than the observation space, which is possible as latents map to distinct regions of the world following the single-parent assumption. We show interventions on the latent variable that controls equatorial Pacific temperatures, and the direct correspondence between latent-space values and observation-space temperatures (Figure 26). Unlike traditional models, generating multiple counterfactual projections is simple and rapid in our framework.

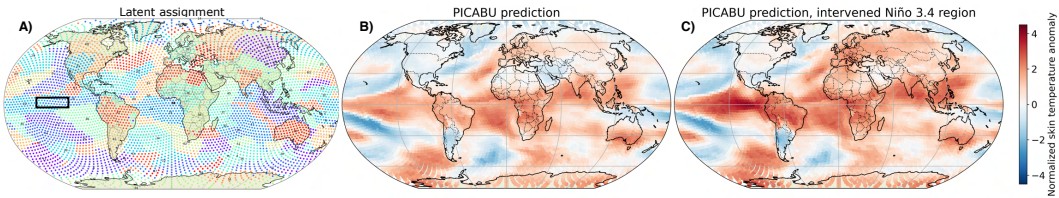

Figure 5: **Intervention on the grid cells that describe the ENSO state.** The left panel shows mapping from latents to observations (colored grid points), and the intervened grid cells. We increase their value, which then influences the latents at the current and next timestep through the learned causal graph. No other grid cells or latent variables are intervened on. The middle panel shows the original, unintervened next-step prediction, and the right panel shows the intervened prediction.

## 5  Discussion

We present a novel approach for interpretable climate emulation, developing a causal representation learning model to identify internal causal drivers of climate variability and generate physically consistent outputs over climate timescales. By propagating probabilistic predictions through the Bayesian filter, our model provides a distribution over projections and encodes uncertainty. The causal nature of the model allows for counterfactual experiments that uncover the drivers of large-scale climate events and aids generalization to unseen climate scenarios, which is critical if emulators are used to explore different emission scenarios.

**Limitations and future work.** The main limitation of our model arises from the single-parent decoding assumption, which effectively partitions the observation space into subspaces mapped to each latent. A high latent dimension is required to represent certain complex prediction spaces, which may contribute to the high-frequency noise we observe in GMST. While single-parent decoding remains a necessary assumption to enable the recovery of causal structure under our framework, relaxing this assumption would be a promising area of future inquiry. On the climate science side, considering anthropogenic emissions and allowing the model to represent connections between the different variables in the climate system are natural next steps for improved emulation of climate model data under different emission scenarios. Training the model to attribute present-day extreme events to climate change (using reanalysis data) is also a promising avenue for future work.

# 6 Acknowledgements

This project was supported by the Intel-Mila partnership program, IVADO, Canada CIFAR AI Chairs program, and Canada First Research Excellence Fund. We acknowledge computational support from Mila – Quebec AI Institute, including in-kind support from Nvidia Corporation. S.H. was supported by funding from EPSRC (EP/S022961/1). P.N. was supported by the UK Natural Environment Research Council (Grant NE/V012045/1). A.T.A. was financially supported by NERC through NCAS (R8/H12/83/003). The authors thank Charlotte E.E. Lange for helping to build the competing methods.

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

## A   Data availability

Processed data used in this work are available at a Zenodo repository (https://zenodo.org/records/14773929). Raw NorESM2 data can be downloaded from the ESGF CMIP6 data store.

## B   Causal Discovery with Single-Parent Decoding

### B.1   CDSD generative model

CDSD considers a generative model where $d_x$-dimensional variables $\{\mathbf{x}^t\}_{t=1}^T$ are observed across $T$ time steps. These observations, $\mathbf{x}^t$, are influenced by $d_z$-dimensional latent variables $\mathbf{z}^t$. For instance, $\mathbf{x}^t$ could represent climate measurements, while $\mathbf{z}^t$ might represent unknown regional climate trends.

The model considers a stationary time series of order $\tau$ over these latent variables. Binary matrices $\{G^k\}_{k=1}^{\tau}$ represent the causal relationships between latents at different time steps. Specifically, an element $G_{ij}^k = 1$ indicates that $z_j^{t-k}$ is a causal parent of $z_i^t$, capturing the lagged causal relations between the time-steps $t-k$ and $t$. The adjacency matrix $F$ delineates the causal connections between the latent variables $\mathbf{z}$ and the observed variables $\mathbf{x}$. Each observed variable $x_i$ has at most one latent parent, adhering to the *single-parent decoding* structure.

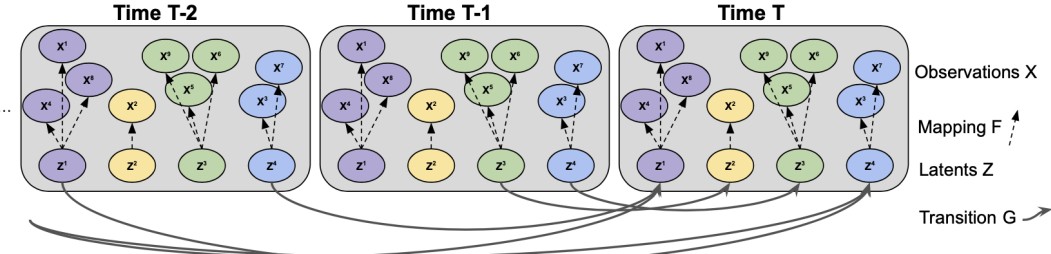

Figure 6: **Generative model.** Variables $\mathbf{z}$ are latent, and $\mathbf{x}$ are observable. $G$ (full arrows) represents latent connections across different time lags. $F$ (dashed arrows) connects latents to observables.

At any given time step $t$, the latents are assumed to be independent given their past, and each conditional is parameterized by a non-linear function $g_j$. $h$ is chosen to be a Gaussian density function.

$$p(\mathbf{z}^t \mid \mathbf{z}^{t-1}, \dots, \mathbf{z}^{t-\tau}) := \prod_{j=1}^{d_z} p(z_j^t \mid \mathbf{z}^{t-1}, \dots, \mathbf{z}^{t-\tau}); \tag{6}$$

$$p(z_j^t \mid \mathbf{z}^{<t}) := h(z_j^t; \ g_j([G_{j:}^1 \odot \mathbf{z}^{t-1}, \dots, G_{j:}^\tau \odot \mathbf{z}^{t-\tau}])), \tag{7}$$

The observable variables $x_j^t$ are assumed to be conditionally independent where $f_j : \mathbb{R} \to \mathbb{R}$, and $\sigma^2 \in \mathbb{R}_{>0}^{d_x}$ are decoding functions:

$$p(x_j^t \mid \mathbf{z}_{pa_j^F}^t) := \mathcal{N}(x_j^t; f_j(\mathbf{z}_{pa_j^F}^t), \sigma_j^2), \tag{8}$$

The model's complete density is:

$$p(\mathbf{x}^{\leq T}, \mathbf{z}^{\leq T}) := \prod_{t=1}^T p(\mathbf{z}^t \mid \mathbf{z}^{<t}) p(\mathbf{x}^t \mid \mathbf{z}^t). \tag{9}$$

Maximizing $p(\mathbf{x}^{\leq T}) = \int p(\mathbf{x}^{\leq T}, \mathbf{z}^{\leq T}) \, d\mathbf{z}^{\leq T}$ unfortunately involves an intractable integral, hence the model is fit by maximizing an evidence lower bound (ELBO) [Kingma and Welling, 2014,

Girin et al., 2021] for $p(\mathbf{x}^{\leq T})$. The variational approximation of the posterior $p(\mathbf{z}^{\leq T} \mid \mathbf{x}^{\leq T})$ is $q(\mathbf{z}^{\leq T} \mid \mathbf{x}^{\leq T})$.

$$q(\mathbf{z}^{\leq T} \mid \mathbf{x}^{\leq T}) := \prod_{t=1}^{T} q(\mathbf{z}^t \mid \mathbf{x}^t); \quad q(\mathbf{z}^t \mid \mathbf{x}^t) := \mathcal{N}(\mathbf{z}^t; \tilde{\mathbf{f}}(\mathbf{x}^t), \mathrm{diag}(\tilde{\sigma}^2)), \tag{10}$$

$$\log p(\mathbf{x}^{\leq T}) \geq \sum_{t=1}^{T} \Big[ \mathbb{E}_{\mathbf{z}^t \sim q(\mathbf{z}^t \mid \mathbf{x}^t)} \big[ \log p(\mathbf{x}^t \mid \mathbf{z}^t) \big] - \mathbb{E}_{\mathbf{z}^{<t} \sim q(\mathbf{z}^{<t} \mid \mathbf{x}^{<t})} \mathrm{KL} \big[ q(\mathbf{z}^t \mid \mathbf{x}^t) \,||\, p(\mathbf{z}^t \mid \mathbf{z}^{<t}) \big] \Big]. \tag{11}$$

The graph between the latent $\mathbf{z}$ and the observable $\mathbf{x}$ is parameterized using a weighted adjacency matrix $W$. To enforce the single-parent decoding, $W$ is constrained to be non-negative with orthonormal columns. Neural networks are optionally used to parameterize encoding and decoding functions $g_j, f_j, \tilde{\mathbf{f}}$. The graphs $G^k$ are sampled from $G_{ij}^k \sim Bernoulli(\sigma(\Gamma_{ij}^k))$, with $\Gamma^k$ being learnable parameters. The objective is optimized using stochastic gradient descent, leveraging the Straight-Through Gumbel estimator [Jang et al., 2017] and the reparameterization trick [Kingma and Welling, 2014].

## B.2   Inference with CDSD: Objective and Optimization

In this section, we present how inference and optimization are carried out when using CDSD [Brouillard et al., 2024].

**Continuous optimization.** The graphs $G^k$ are learned via continuous optimization. They are sampled from distributions parameterized by $\Gamma^k \in \mathbb{R}^{d_z \times d_z}$ that are learnable parameters. Specifically, $G_{ij}^k \sim Bernoulli(\sigma(\Gamma_{ij}^k))$, where $\sigma(\cdot)$ is the sigmoid function. This results in the following constrained optimization problem, with $\phi$ denoting the parameters of all neural networks ($r_j, g_j, \tilde{\mathbf{f}}$) and the learnable variance terms at Equations 8 and 10:

$$\max_{W, \Gamma, \phi} \mathbb{E}_{G \sim \sigma(\Gamma)} \big[ \mathbb{E}_{\mathbf{x}} \left[ \mathcal{L}_{\mathbf{x}}(W, \Gamma, \phi) \right] \big] - \lambda_s ||\sigma(\Gamma)||_1$$
$$\text{s.t. } W \text{ is orthogonal and non-negative,} \tag{12}$$

$\mathcal{L}_{\mathbf{x}}$ is the ELBO corresponding to the right-hand side term in Equation (11) and $\lambda_s > 0$ a coefficient for the regularization of the graph sparsity. The non-negativity of $W$ is enforced using the projected gradient on $\mathbb{R}_{\geq 0}$, and its orthogonality enforced using the following constraint:

$$h(W) := W^T W - I_{d_z} .$$

This results in the final constrained optimization problem, relaxed using the *augmented Lagrangian method* (ALM):

$$\max_{W, \Gamma, \phi} \mathbb{E}_{G \sim \sigma(\Gamma)} \big[ \mathbb{E}_{\mathbf{x}}[\mathcal{L}_{\mathbf{x}}(W, \Gamma, \phi)] \big] - \lambda_s ||\sigma(\Gamma)||_1 - \mathrm{Tr}\left( \lambda_W^T h(W) \right) - \frac{\mu_W}{2} ||h(W)||_2^2, \tag{13}$$

where $\lambda_W \in \mathbb{R}^{d_z \times d_z}$ and $\mu_W \in \mathbb{R}_{>0}$ are the coefficients of the ALM.

This objective is optimized using stochastic gradient descent. The gradients w.r.t. the parameters $\Gamma$ are estimated using the Straight-Through Gumbel estimator [Jang et al., 2017]. The ELBO is optimized following the classical VAE models [Kingma and Welling, 2014], by using the reparametrization trick and a closed-form expression for the KL divergence term since both $q(\mathbf{z}^t \mid \mathbf{x}^t)$ and $p(\mathbf{z}^t \mid \mathbf{z}^{<t})$ are multivariate Gaussians. The graphs $G$ and the matrix $W$ are thus learned end-to-end.

## B.3   Parameter sharing

The original model parameterizes the distribution of the observations with one neural network per grid point and variables, and the transition model with one neural network per latent variable, leading to a linear increase in neural networks with the number of latents and input dimensions. We implement parameter sharing for computational efficiency. More precisely, there is now a single neural network to parameterize the mapping and a single neural network to parameterize the transition model. The

neural networks take as input a positional embedding that is learned through training, and represents each grid location and input variable (non-linear mapping from latents to observations), and each latent (non-linear transition model).

# C    SAVAR datasets

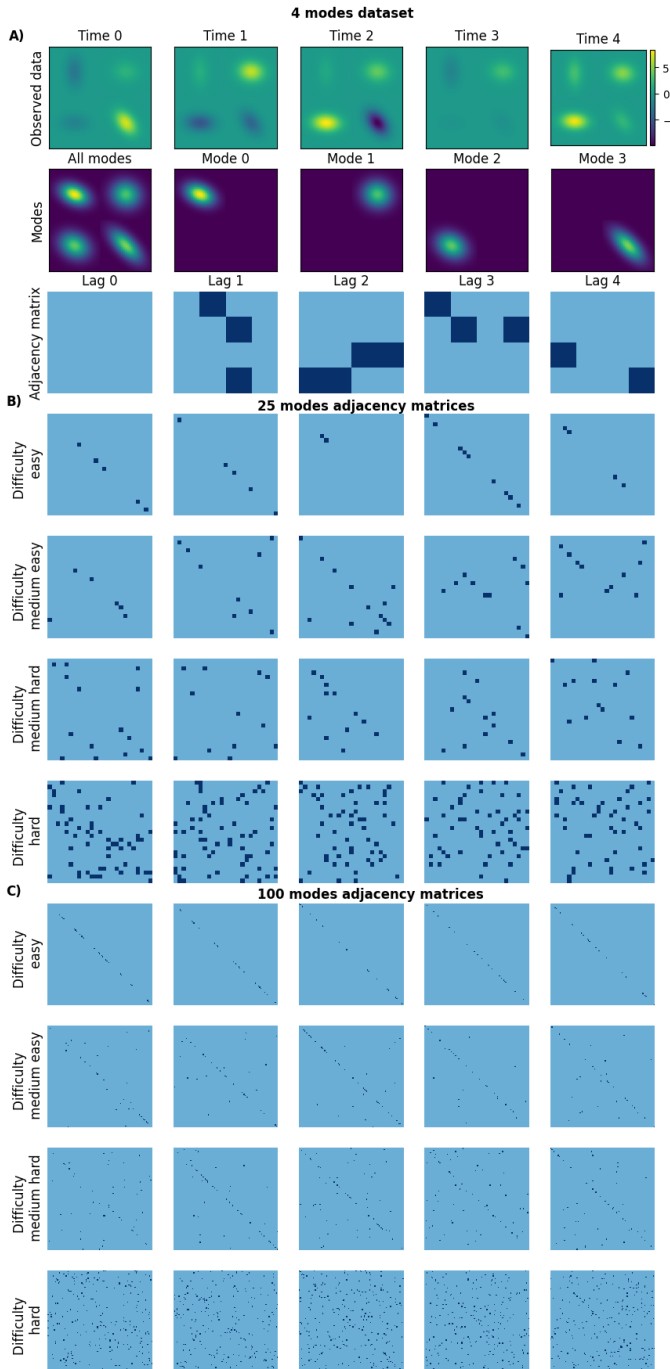

Figure 7: **SAVAR datasets with 4, 25, and 100 latent dimensions.** A) An example SAVAR dataset generated with four latent dimensions. The first row shows five timesteps of observed data; the second shows the four modes of variability, each corresponding to a latent, and the third row shows the adjacency matrix representing the connections between the latents at time t and the five previous timesteps. B) and C) show, respectively, for the 25 and 100 latent datasets, the adjacency matrix corresponding to the four levels of difficulty.

## C.1 Distribution of links strengths

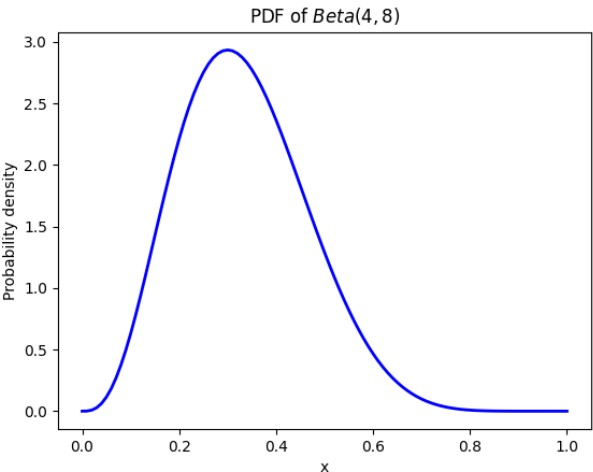

Figure 8: **Probability distribution function of the distribution of SAVAR link strengths.** The beta distribution $\mathcal{B}(4, 8)$ was chosen to have mostly moderate link strengths, centered around $1/3$, with few values very close to 0 and 1 and relatively low standard deviation ($0.13$).

## C.2 Details on the training procedure for all models

Here, we give training details for Varimax-PCMCI, CDSD, and PICABU.

PICABU was trained using the parameters shown in Table 7. The only difference was the sparsity constraint "Constrained value", which was set to match the expectation of the causal graph, as explained in Section 4.1.

CDSD was trained using the same parameters, except that instead of a sparsity constraint, it uses a penalty. The coefficient $\mu$ was thus fixed. We performed a hyperparameter search over $\mu$ and tested values $10^{-16}, 10^{-14}, ..., 10^2$. We retained the value leading to the highest F1 score. Moreover, we stopped the training when the sparsity was reaching its expected value.

PCMCI was trained using ParCorr to perform the conditional independence tests, and the significance value of the tests was replaced to keep the $M$ most significant instead, where $M$ is the expected number of edges in the graph.

An advantage of PICABU is the possibility to control the final sparsity of the graph directly, as it is optimized using a constraint. We thus set the sparsity equal to the expected number of edges in the true graph $M$. In comparison, CDSD utilizes a penalty to optimize for sparsity. One drawback is that if the penalty coefficient is too high or training lasts too long, the learned graph will be too sparse. For fair comparison, we stop the training of CDSD when the number of connections of strength at least $0.5$ is equal to $M$. We train with various values for the sparsity penalty coefficient, and report the maximum F1 score obtained. PCMCI instead computes conditional independence tests between all possible pairs of latent variables, and keeps edges when the test rejects conditional independence. The test is parameterized by a threshold for p-values: if the test's p-value is above the threshold, conditional independence is rejected. For fair comparison, we modify its procedure to keep links for the $M$ conditional independence tests with the highest p-values. The final parameters used for the three methods are given in Appendix C.

## C.3 Inferring causally-relevant latents improves accuracy

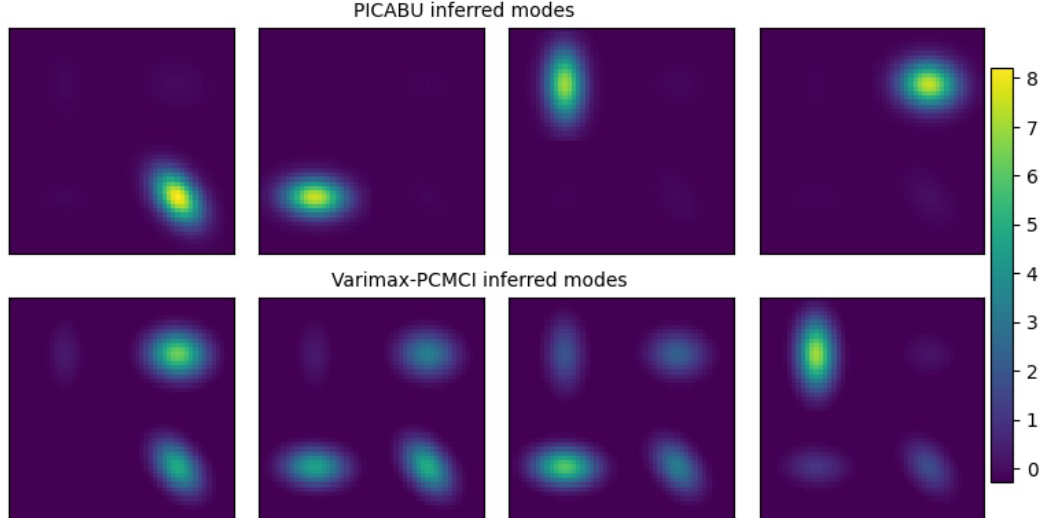

Figure 9: **PICABU disentangles latents well.** The top row shows the four modes of variability identified by PICABU, in the dataset shown in Figure 7 (A). The bottom row shows the four modes identified by Varimax-PCMCI. The latents are not very well disentangled, as they are inferred independently of the causal graph, and PCA exhibits timescale mixing and exhibit timescale mixing Aubry et al. [1993]. Varimax-PCA is not able to disentangle them because of the autoregressive dependencies present in the data.

## D  Training details for ViT + positional encoding model

The ViT + pos. encoding model is adapted from ClimaX [Nguyen et al., 2023]. This architecture is originally designed to take multiple variables as input. We adapted it to only emulate temperature. We trained it with various hyperparameters and report results with the best test loss. We tried different network depths (from 1 to 8) and got best results with 2, various learning rates (from 0.00001 to 0.1) and got best results with 0.0001, various batch sizes (from 16 to 512) and got best results with 64. The best architecture is significantly smaller than the original one since it is trained to emulate one variable. Moreover, it is trained to predict multiple timesteps into the future by having a fixed encoder and multiple prediction heads, each corresponding to a different prediction horizon. Here, we train it to predict 1, 2, 3, 4, or 5 timesteps into the future to compare to PICABU with $\tau = 5$. During rollouts, we select the prediction head corresponding to $\tau = 1$ to emulate monthly trajectories.

The architecture is adapted from an approach designed to be trained with more data sources and input variables, and at multiple time resolutions. We find that when trained on a smaller dataset, it is unable to learn the dynamics of the system directly. The architecture is trained with a latitude-weighted mean squared error [Rasp et al., 2020], which will suffer from learning low spatial frequencies. Hence, the model has low bias but very high range and variability.

## E  Example time series of NorESM2 data

Figure 10 shows an example time series of NorESM2 data, illustrating six consecutive months of data. These data have been normalized and deseasonalized. This time series illustrates important features of seasonal and climate variability. For example, while the Pacific Ocean shows relatively unchanging temperatures, temperatures in the extratropics are much more variable and less predictable. Note that for our experiments on NorESM2 data, PICABU takes as input the previous five months (here, $t - 5$ to $t - 1$), and then learns to predict timestep $t$ based on these past timesteps.

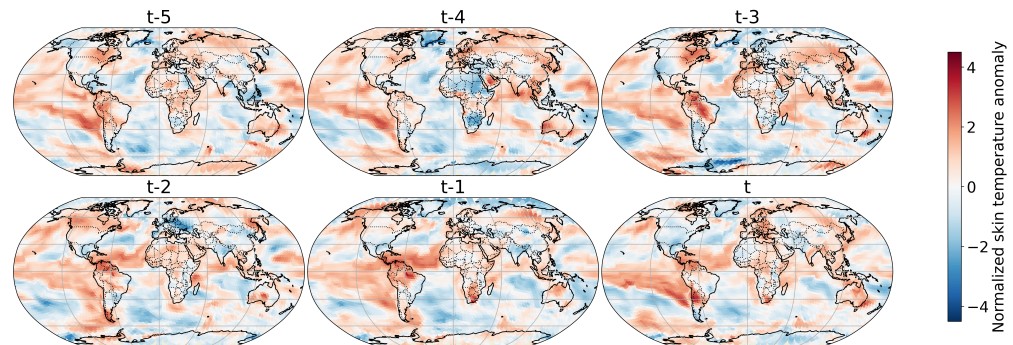

Figure 10: **Example time series of six months of NorESM2 data.** Here we illustrate an example of a 6-month time series of NorESM2 data. PICABU takes as input the previous five months, and then aims to predict timestep $t$.

# F Further climate emulation results

To describe the performance of PICABU for long rollouts, we illustrate an example time series of the Niño3.4 index over a 50-year rollout, and similarly, we show the time series of normalized global mean surface temperature in Figure 11. These rollouts are generated from a randomly chosen initial condition. The ENSO variability generated by the emulator is realistic, capturing the distribution of observed variability well without drift. While the GMST similarly does not drift away from the expected mean value, our emulator exhibits too much variance over short timescales.

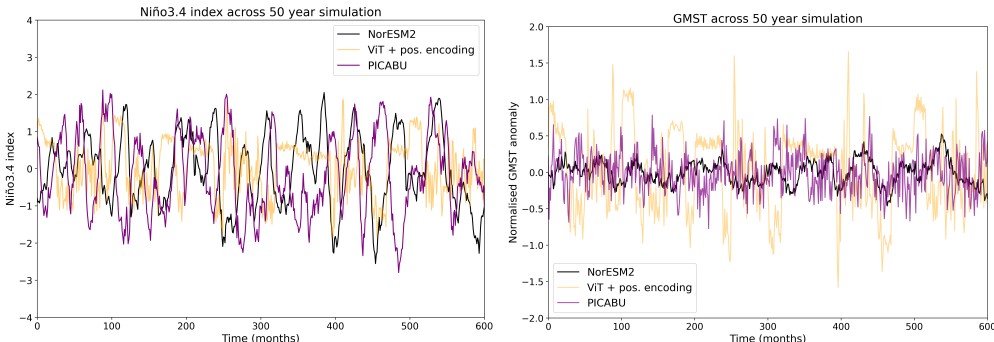

Figure 11: **PICABU generates realistic ENSO variability, but shows too much short-term variability in GMST**. We compare ENSO, as measured by the Niño3.4 index, and GMST for the ground truth NorESM2, the ViT architecture, and PICABU over 50 years. While PICABU captures ENSO variability well and generates more realistic variability compared to the ViT model, it fails to represent the short-term variability of GMST accurately.

This observation is confirmed by the power spectra for ENSO and GMST. While the ENSO power spectrum generated by PICABU is accurate at both low and high frequencies, the power for GMST is too high at high frequencies, as shown in Figure 4. This is true across all variants of PICABU that we train. The V-PCMCI model, however, has a more accurate spectrum at high frequencies but very little power at low frequencies. We also show the spectra at higher frequencies, on a log scale in Figure 12, where it is clear that PICABU captures the high frequencies of ENSO much better than for GMST.

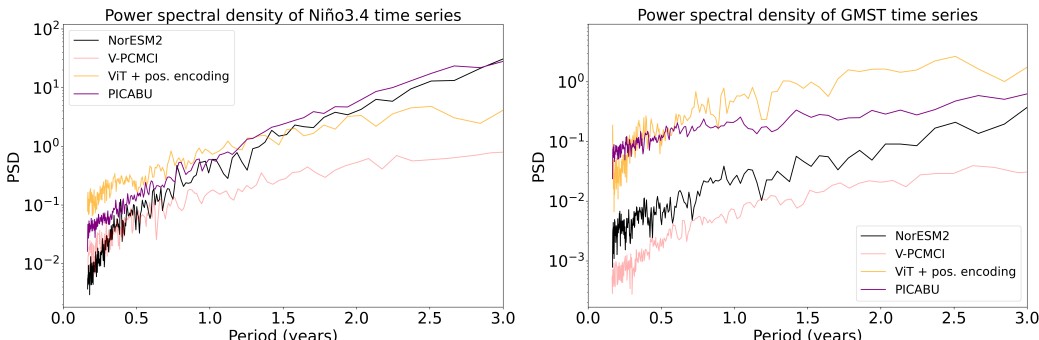

Figure 12: **PICABU captures the high-frequencies of ENSO well, but not of GMST.** The power spectra for ENSO and GMST, focused on the high frequencies, and plotted on a log scale, showing that PICABU performs better than competing methods, but has too much power at very high frequencies.

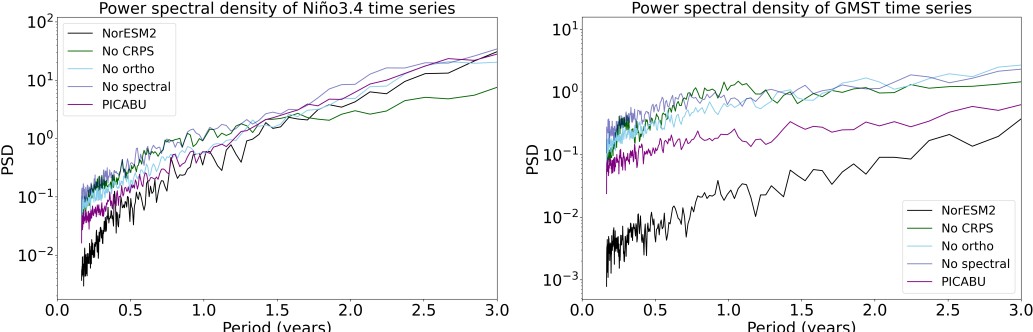

Figure 13: **PICABU captures the power spectrum of both ENSO and GMST better than ablated models.** The power spectra for ENSO and GMST, focused on the high frequencies, and plotted on a log scale, showing that PICABU better approximates the ground truth compared to the ablated models.

## G Annual temperatures ranges generated by PICABU

Here we illustrate the average intra-annual temperature range for NorESM2 and PICABU simulations over a 100-year simulation. For each grid cell, for each year, we take the month with the maximum value, and the month with the minimum value, and then compute the difference to give the intra-annual range. This quantity is then averaged over the 100 years of simulation for both NorESM2 and PICABU to generate Figure 14. The spatial distribution of the ranges is reasonable, with some notable exceptions over land (such as sub-Saharan Africa), where the range is much too small. The ranges over the ocean largely agree.

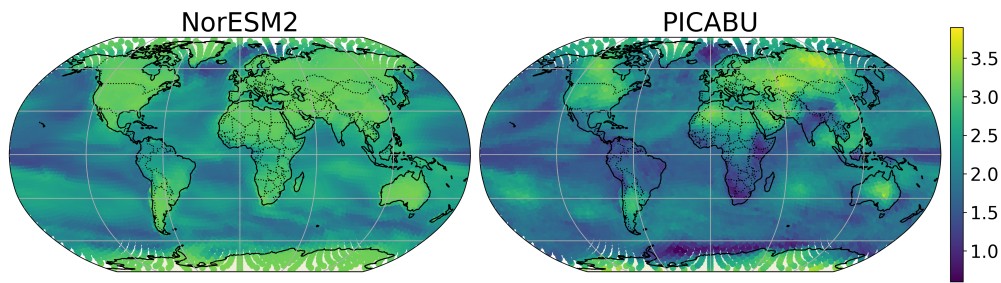

Figure 14: **Annual temperature range of NorESM2 and PICABU.** The range is defined as the intra-annual range of temperatures at each grid cell, averaged over a 100-year simulation. While the spatial distribution of the temperature ranges agrees well, the emulator exhibits a greater intra-annual range in temperature than the climate model.

## H Indian Ocean Dipole and Atlantic Multidecadal Oscillation emulation

We also evaluated the performance of PICABU in emulating the variability of two other major modes of climate variability, the Indian Ocean Dipole (IOD) and the Atlantic Meridional Oscillation (AMO), which have wide-ranging effects on climate variability across the globe [Saji et al., 1999, Kerr, 2000, McCarthy et al., 2015]. The IOD is determined by fluctuations in sea surface temperatures, where the western Indian Ocean is warmer or colder (positive and negative phases) than the eastern Indian Ocean. The index is the monthly difference in temperature anomalies between the western (10°S - 10°N, 50° - 70°E) and eastern parts of the Indian Ocean (10°S - 0°, 90° - 110°E). The Atlantic Multidecadal Oscillation is calculated as the anomaly in sea surface temperatures in the North Atlantic, typically between 0° and 60°N.

Figure 15 shows example time series of NorESM's simulated IOD and AMO, alongside PICABU's emulation, and also shows the power spectra of the time series of both modes of variability. The quantitative performance is given in Table 4.

The performance of PICABU shows a similar trend to the performance for ENSO and GMST. More predictable, smaller spatial-scale variability, in this case IOD variability, is modeled well, with closely matching power spectra and realistic time series. However, the larger-scale feature (in both time and space), AMO, is less accurately emulated, with too much variability at high frequencies, similar to the results for GMST, though more muted.

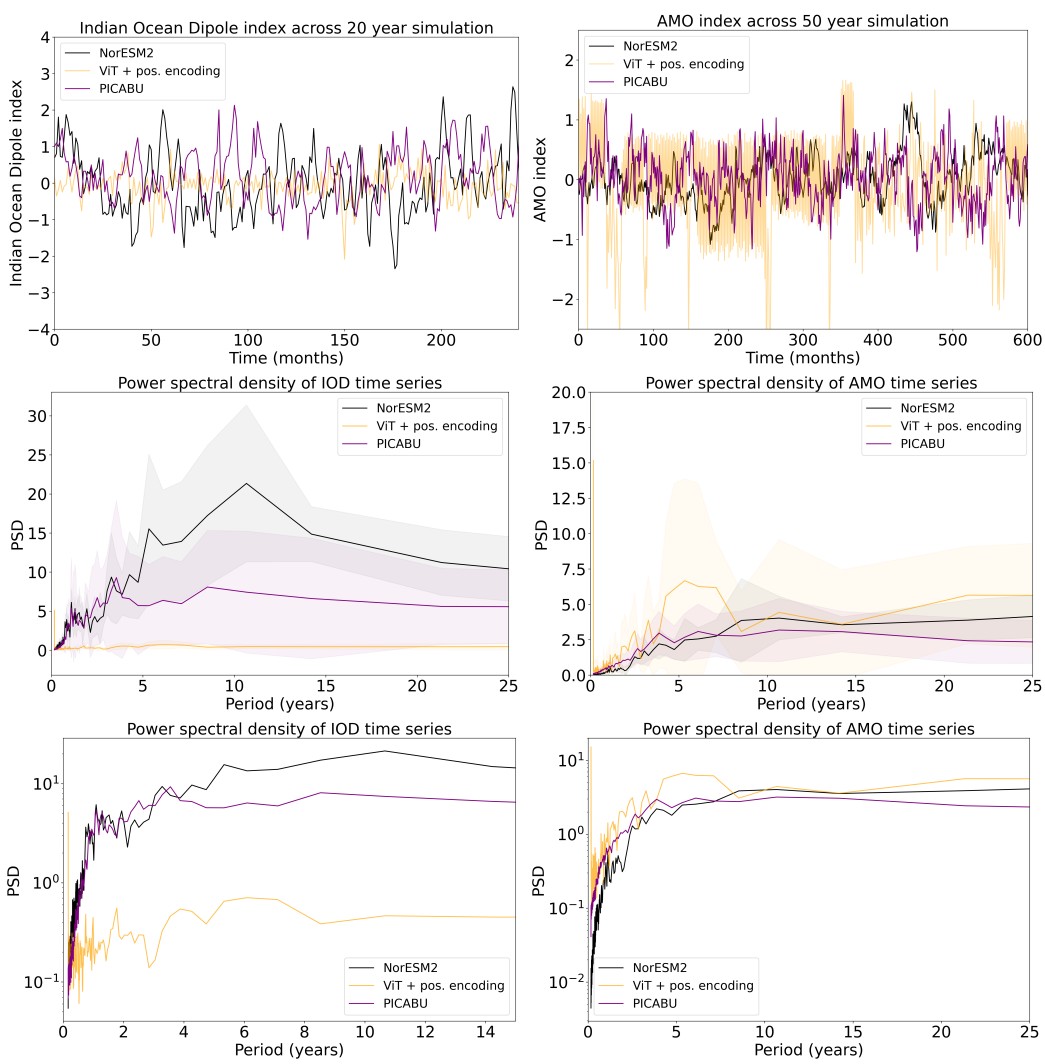

Figure 15: **PICABU generates realistic IOD variability, but does not capture AMO short-term variability**. The figure shows example time series of the IOD and AMO for the ground truth climate model NorESM2, the ViT model, and PICABU. The power spectra of the time series, including on a log scale, are also shown.

|  |  | **Mean** | **Std. Dev.** | **Range** | **LSD** |
|---|---|---|---|---|---|
| **IOD** | ViT + pos. encod. | 0.324 | 0.231 | 3.11 | 0.343 |
|  | PICABU | 0.186 | 0.753 | 5.36 | 0.0772 |
|  | Ground truth | -8.44e-8 | 0.877 | 7.36 | Truth |
| **AMO** | ViT + pos. encod. | 0.187 | 0.897 | 4.17 | 0.712 |
|  | PICABU | 0.0538 | 0.445 | 3.27 | 0.248 |
|  | Ground truth | -7.11e-8 | 0.368 | 2.58 | Truth |

Table 4: **Evaluation of PICABU for IOD and AMO emulation of NorESM2.** The results show the quantitative performance of PICABU in simulating IOD and AMO variability in NorESM2, across five 100 year simulations for both NorESM2 and PICABU.

# I Results of CESM2-FV2 emulation

In addition to training and evaluating PICABU for NorESM2 emulation, we also separately trained and evaluated on a second climate model, CESM2-FV2. We did this to investigate whether our model is able to emulate another climate model, how sensitive PICABU's training is to hyperparameters, and whether our findings regarding PICABU's performance are consistent for another climate model. It should be noted that NorESM2 is based on CESM2 and, therefore, has many shared model components. NorESM2 has a different model for the ocean, ocean biogeochemistry, and atmospheric aerosols. It also includes specific modifications and tunings of the dynamics of the atmosphere model [Seland et al., 2020]. However, the two models are not completely independent.

With only minimal hyperparameter tuning, we train PICABU on CESM2-FV2, and illustrate comparable performance as when emulating NorESM2. While training with the same hyperparameters as for NorESM2 produced stable rollouts immediately, we improved performance by decreasing one hyperparameter, the spatial spectral coefficient, by 50%. This shows that PICABU is robust and can be trained with minimal hyperparameter tuning to easily emulate different climate models.

In Figure 16 we illustrate example time series of global mean surface temperature (GMST), Niño3.4, the IOD, and the AMO, as also shown for NorESM2, giving a qualitative illustration of the variability generated by PICABU for each. Figure 17 and Figure 18 show the power spectra of PICABU and the ground truth CESM2 model for the different indices, showing the spectra on a standard and log plot, respectively. The quantitative results are presented in Table 5.

The results are similar to PICABU's performance in emulating NorESM2 data. The modes with shorter-term variability are modeled well, capturing relevant peaks for Niño3.4 and the IOD, while the temporal variability at larger scales is also not captured accurately.

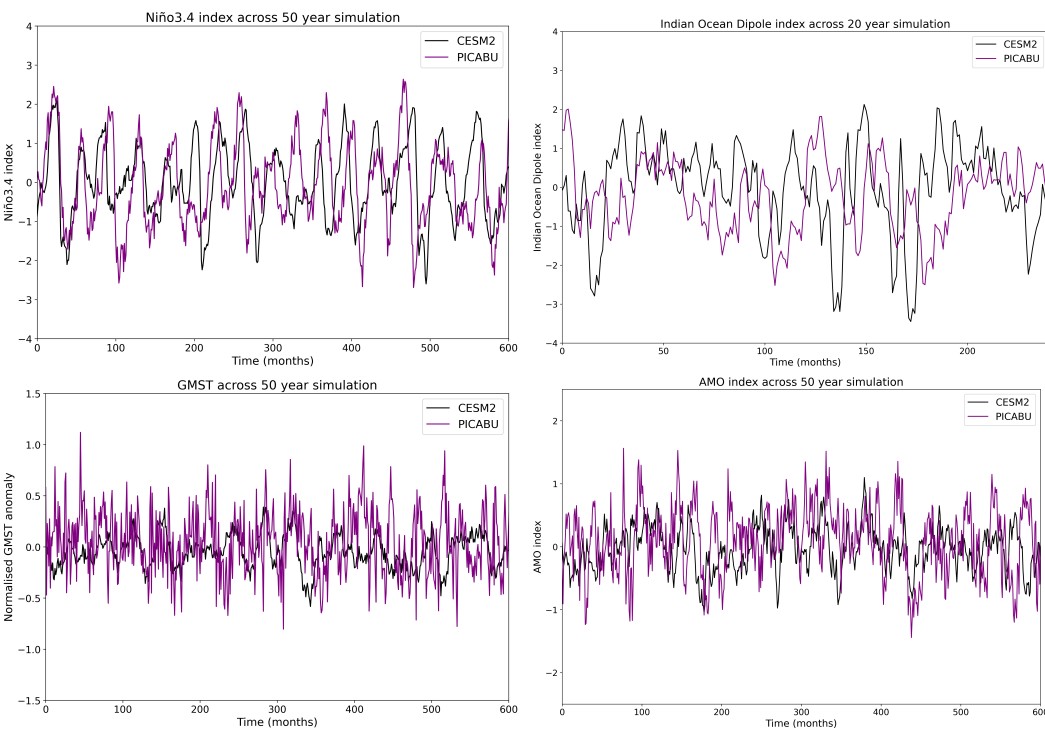

Figure 16: **For CESM2-FV2, similar to NorESM2, PICABU generates realistic ENSO and IOD variability, but does not capture GMST short-term variability**. We provide example time series of the Niño3.4 index and GMST for both the ground truth climate model CESM2-FV2 and PICABU over 50 years. Example time series of the IOD and AMO are also shown.

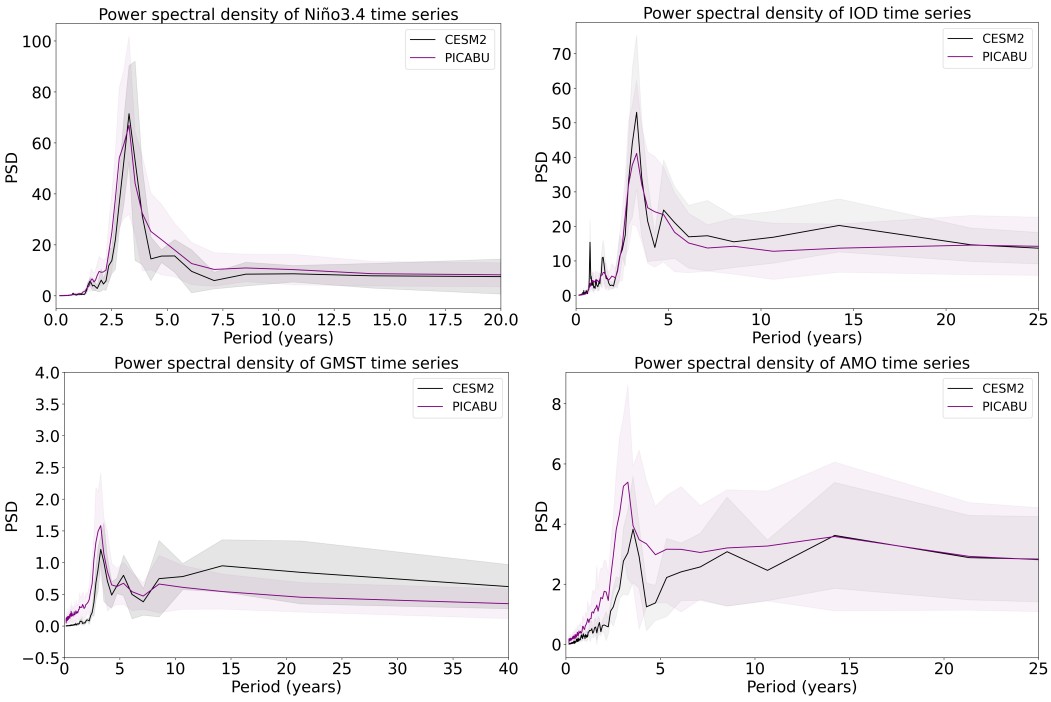

Figure 17: **CESM2-FV2 emulation results are consistent with the NorESM2 emulation results.** The power spectra for four measures of climate variability, the Niño 3.4 index, IOD, GMST, and AMO are shown, both for the ground truth CESM2-FV2 model and for rollouts generated by PICABU.

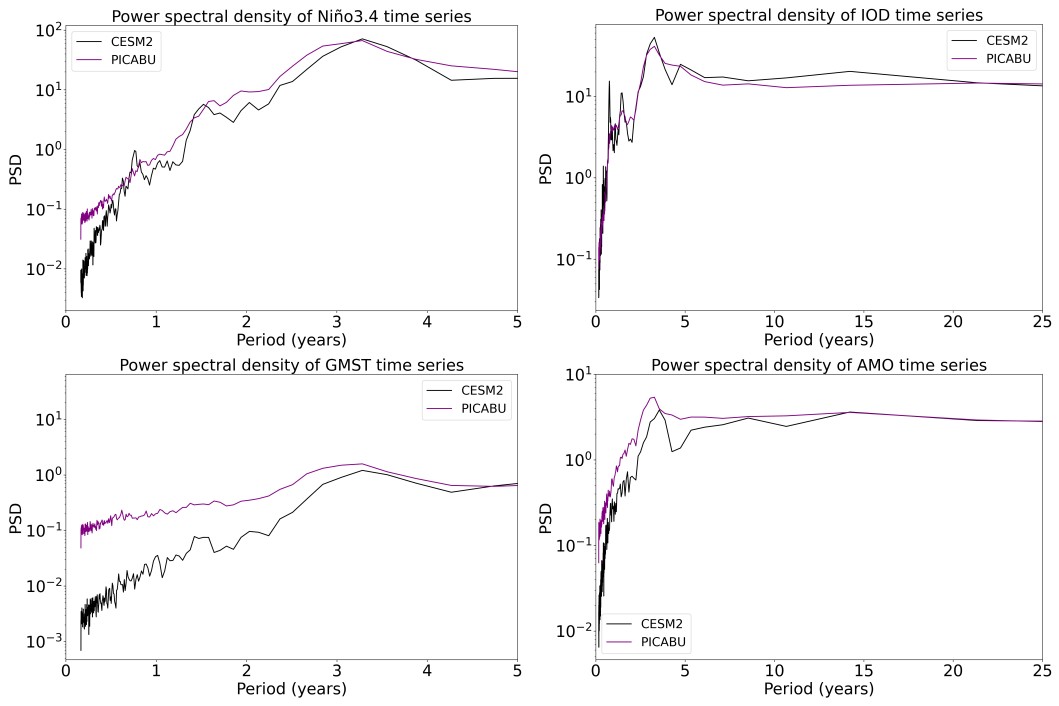

Figure 18: **PICABU predictions exhibit too much power at high frequencies.** The power spectra on a log scale for four measures of climate variability, ENSO, IOD, GMST, and AMO are shown, both for the ground truth CESM2-FV2 model, and rollouts generated by PICABU.

|  |  | **Mean** | **Std. Dev.** | **Range** | **LSD** |
|---|---|---|---|---|---|
| **GMST** | PICABU | 0.0407 | 0.310 | 2.60 | 0.493 |
|  | Ground truth | 0 | 0.171 | 1.30 | 0 |
| **Niño3.4** | PICABU | -0.0182 | 1.10 | 7.57 | 0.231 |
|  | Ground truth | 0 | 0.954 | 5.22 | 0 |
| **IOD** | PICABU | -0.424 | 1.12 | 7.56 | 0.0729 |
|  | Ground truth | 0 | 1.15 | 8.50 | 0 |
| **AMO** | PICABU | 0.138 | 0.503 | 3.81 | 0.265 |
|  | Ground truth | 0 | 0.377 | 2.76 | 0 |

Table 5: **Evaluation of PICABU for CESM2-FV2 climate model emulation.** The results show the quantitative performance of PICABU in simulating various modes of variability in the CESM2-FV2 climate model.

## J   Choice of constant variance in the Bayesian filter

When running a standard Bayesian filter, the variance for the distribution of the predicted samples is typically the variance of the distribution observed in the model. Table 2 shows that PICABU learns a distribution with higher variance than the observations. To avoid propagating this higher uncertainty through the prediction, we instead use the assumption of constant variance, which is well-suited to the pre-industrial control data, and estimate this constant variance from the observations directly. Table 6 shows summary statistics when using constant variance vs. the model's inferred variance.

| | Mean | Std. Dev. | Range | LSD | Mean | Std. Dev. | Range | LSD |
|---|---|---|---|---|---|---|---|---|
| | **GMST** | | | | **Niño3.4** | | | |
| Ground truth | 0 | 0.177 | 1.357 | 0 | 0 | 0.927 | 5.73 | 0 |
| Constant var. | -0.0518 | 0.277 | 2.02 | 0.444 | -0.0914 | 1.05 | 5.99 | 0.191 |
| Inferred var. | -0.0742 | 0.324 | 2.15 | 0.464 | -0.198 | 1.27 | 7.70 | 0.206 |
| | **IOD** | | | | **AMO** | | | |
| Ground truth | 0 | 0.877 | 7.36 | 0 | 0 | 0.368 | 2.58 | 0 |
| Constant var. | 0.186 | 0.753 | 5.36 | 0.0772 | 0.0538 | 0.445 | 3.27 | 0.248 |
| Inferred var. | 0.110 | 0.845 | 5.46 | 0.0753 | 0.0157 | 0.498 | 3.27 | 0.261 |

Table 6: **Using the model's estimated variance in the Bayesian filter slightly degrades the predictions.** The table shows statistics (mean, standard deviation, range, log-spectral distance) for the ground truth, prediction using a constant variance estimated from the observations, and the model's inferred variance. The predictions degrade slightly when using the inferred variance, especially the standard deviation and range, as PICABU tends to overestimate the variance of the data.

## K  Ablation of number of latents and sparsity coefficient

We investigate the performance of PICABU when we vary the number of latent variables. Figure 19 shows the clustering for different numbers of latent variables. Across different numbers of latents, PICABU learns similar clustering, with larger clusters being progressively split up as the number of latents increases, while some prominent clusters remain for most numbers of latents. Figure 20 shows the power spectra for Niño3.4 and GMST for models trained with different numbers of latents. All models are stable, though notably for small numbers of latents, PICABU does not represent the 3-year peak in the ENSO power spectrum.

Similarly, we vary the sparsity coefficient, finding that it does not have a considerable effect on the power spectra of the generated rollouts.

## L  Multivariable emulation

While most results in this work focus on a single variable, skin temperature, we illustrate that multivariable emulation is possible. PICABU can be extended to include multiple variables by learning latents for each individual variable, and allowing all of these latents to influence each other, including across variables (this is increasingly expensive if the number of variables and the number of latents is large). Figure 22 shows two previous timesteps of data, the reconstruction from the model (simply encoding and decoding the given timestep), and the next timestep prediction.

## M  Training objective for non-causal model

In Section 4.3, the following objectives are optimized when training the model without sparsity (Equation (14)) and without orthogonality (Equation (15)), respectively:

$$\mathcal{L}^{\{\lambda,\mu\}}_{\text{no sparsity}} = \text{ELBO} + \mathcal{C}^{\{\lambda,\mu\}}_{\text{single parent}} + \lambda_{\text{CRPS}}\mathcal{L}_{\text{CRPS}} + \lambda_{\text{s}}\mathcal{L}_{\text{spatial}} + \lambda_{\text{t}}\mathcal{L}_{\text{temporal}}, \tag{14}$$

$$\mathcal{L}^{\{\lambda,\mu\}}_{\text{no orthogonality}} = \text{ELBO} + \mathcal{C}^{\{\lambda,\mu\}}_{\text{sparsity}} + \lambda_{\text{CRPS}}\mathcal{L}_{\text{CRPS}} + \lambda_{\text{s}}\mathcal{L}_{\text{spatial}} + \lambda_{\text{t}}\mathcal{L}_{\text{temporal}}, \tag{15}$$

## N  Experiments compute resources

To train PICABU on 800 years of climate model data, we used the following compute resources: 2 RTX8000 GPUs, with 48GB of RAM each, for ≈10 hours. These resources scale linearly with the number of variables, number of latents, or number of input timesteps $\tau$. The synthetic experiments are much faster (≈0.5 hour on 1 RTX8000 GPU), as it is lower dimensional. For our autoregressive

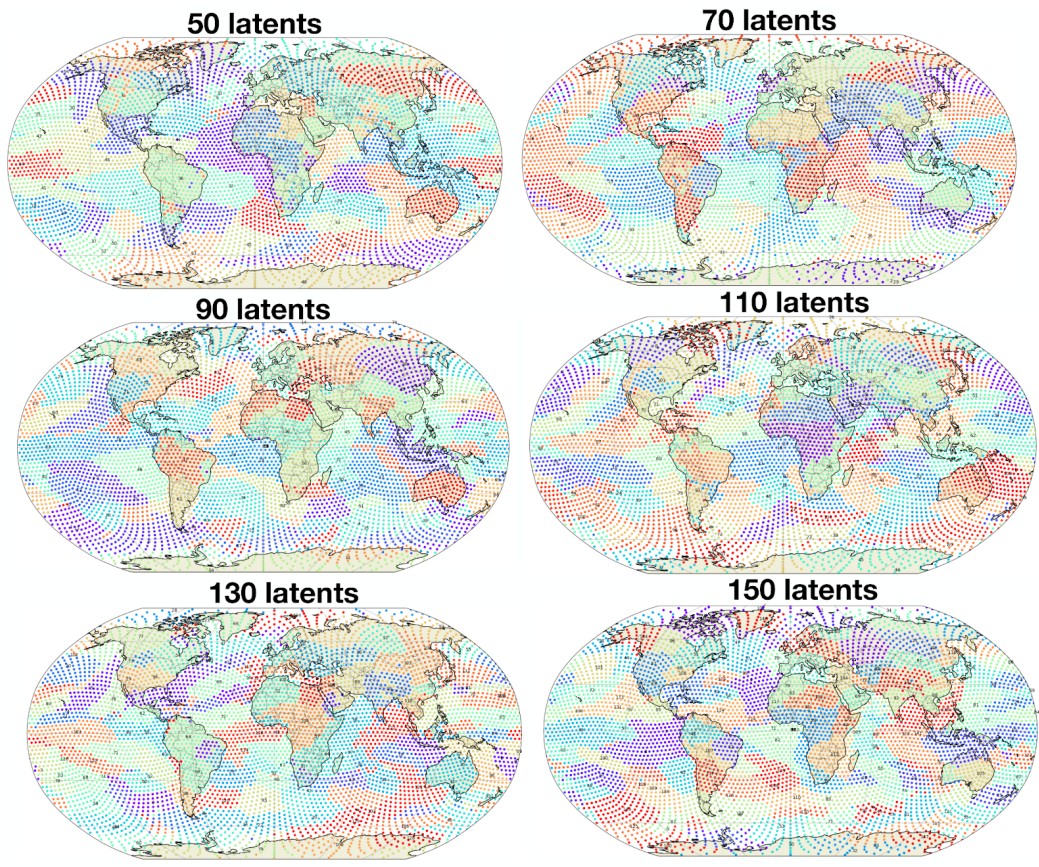

Figure 19: **Spatial aggregation learned by PICABU for different numbers of latent variables.** The spatial aggregation learned by PICABU shows variation as the number of latents is varied. However, some clusters are reasonably consistent across the different numbers of latents.

rollouts emulating climate model data, the use of the Bayesian filter introduces some computational overhead compared to direct deterministic rollouts. However, the operations required for the Bayesian filter (sampling, decoding, and fast Fourier transforms) can be batched at each step and performed rapidly. In general, we are able to simulate 100 years (1200 months) in less than 6 minutes on a single RTX8000 GPU with 48GB memory, with $N = 300$ and $R = 10$ (with the computational time scaling linearly with $N \cdot R$).

All experiments were run on an internal cluster. We conducted hyperparameter tuning on these resources (Appendix O).

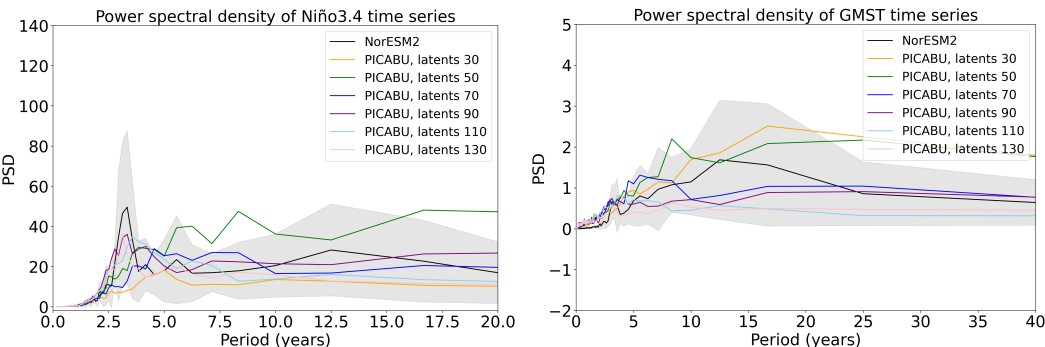

Figure 20: **PICABU is stable when training with different numbers of latent variables.** When varying the number of latent variables, the models are stable and perform well in simulating ENSO, though if the number of latents is small (e.g. 30), the model struggles to represent the 3-year peak in ENSO. Overall, however, this further illustrates that the models are generally stable, not requiring extensive hyperparameter tuning to generate stable rollouts.

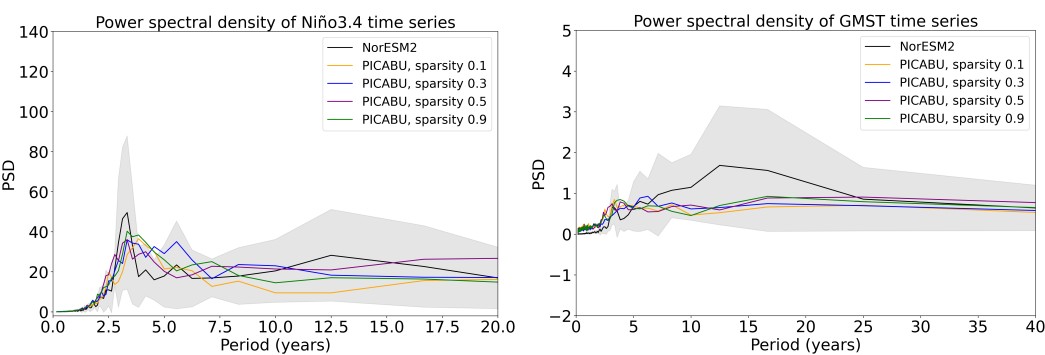

Figure 21: **PICABU is stable when training with different sparsities.** Varying the sparsity constraint produces stable models that perform well in simulating ENSO, keeping all other hyperparameters the same. This illustrates that the models are stable, can be used flexibly, and do not require extensive hyperparameter tuning.

## O  Hyperparameter values

We provide the values of the hyperparameters that we use for the model training and for the Bayesian filtering in Table 7. These hyperparameters were determined with manual tuning, with the values of the spectral penalties being important for model performance. We performed a search over the  20 parameters described in Table 7, with  100 runs of PICABU.

We also found that the initial values of the ALM coefficients were important for effective model training. We empirically observed that if the initial coefficients for the sparsity constraint were too large, the model immediately learned a sparse causal graph before it learned to make accurate predictions or a good latent representation, leading to poor performance. However, if the causal graph was sparsified only later in the training, with smaller initial ALM coefficients, then the model could generate accurate predictions as it first learned causally-relevant latents, only later sparsifying the causal graph to retain important connections.

After the initial set of parameters was selected on NorESM2, we performed minimal parameter tuning to train PICABU on CESM2, as described in Appendix I, and the hyperparameters performed well on the simulated dataset (SAVAR) where only the sparsity of the final graph was changed. These results suggest that in practice, our hyperparameters will provide a sound baseline for adaptation to other

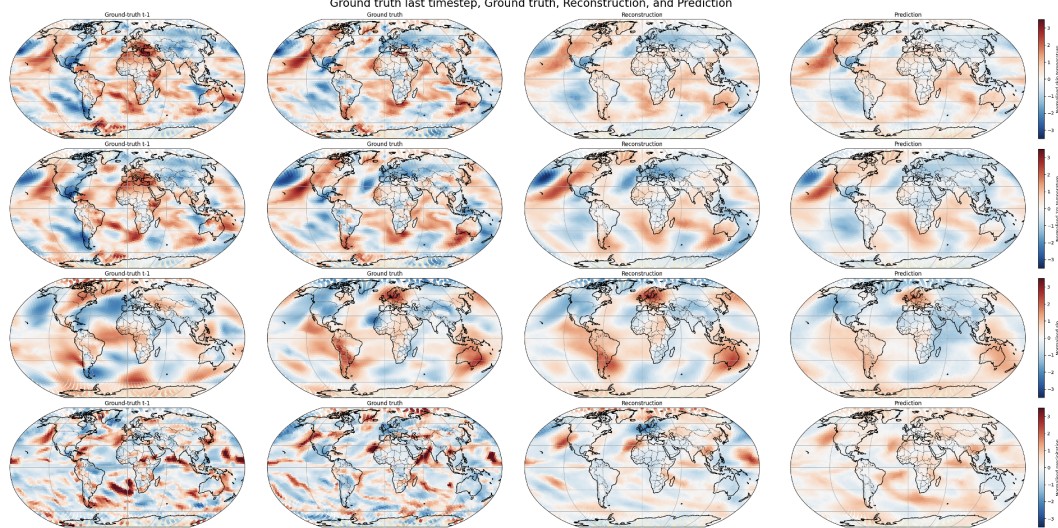

Figure 22: **PICABU may be used for multivariable emulation.** Each row shows a different variable, starting from the top: 1) Skin temperature, 2) surface temperature, 3) sea-level pressure, and 4) precipitation. The first column shows the ground truth at $t - 1$ (previous timestep), and the second column shows the ground truth at time $t$. The third column shows the reconstruction of each variable, and the fourth column shows PICABU's prediction at time $t$.

climate models. Moreover, Appendix M shows that our default hyperparameters are robust while varying other aspects of the model (e.g., number of latents, sparsity).

| | Hyperparameter | Value |
|---|---|---|
| | Learning Rate | 0.0003 |
| | Batch size | 128 |
| | Iterations | 200000 |
| | Optimizer | rmsprop |
| | Number of latents | 90 |
| | $\tau$, number of input timesteps | 5 |
| | CRPS coefficient | 1 |
| | Spatial spectrum coefficient | 3000 |
| | Temporal spectrum coefficient | 2000 |
| **Transition model** | Hidden Layers | 2 |
| | Neurons per Layer | 8 |
| **Encoder-decoder model** | Hidden Layers | 2 |
| | Neurons per Layer | 16 |
| **Sparsity constraint** | Initial $\mu$ | 1e-1 |
| | Multiplication factor $\mu$ | 1.2 |
| | Threshold | 1e-4 |
| | Constrained value | 0.5 |
| **Orthogonality constraint** | Initial $\mu$ | 1e5 |
| | Multiplication factor $\mu$ | 1.2 |
| | Threshold | 1e-4 |
| **Bayesian filtering** | N | 300 |
| | R | 10 |

Table 7: Hyperparameter values used for training PICABU and reported results.

## P  Observation-space interventions

In addition to intervention in observation space on ENSO, we carried out interventions in observation space to perturb the IOD, to explore whether the model responded to these interventions in a manner consistent with known teleconnections (Figure 23). We directly increase the temperature value in the region for the Niño 3.4 intervention, while for the IOD we simply increase the temperature of the western Indian Ocean to increase the IOD index. In line with expected teleconnections, increasing El Niño, as measured by the Niño 3.4 index, causes increased global temperatures with a correlation of 0.9. We also observed, in line with Cai et al. [2009], that in the model there is a causal teleconnection between the springtime strength of the Indian Ocean Dipole and temperatures over Australia, with a correlation of 0.87. Furthermore, interventions over Alaska of a similar magnitude had minimal effects on GMST, with a correlation of 0.08 (not shown).

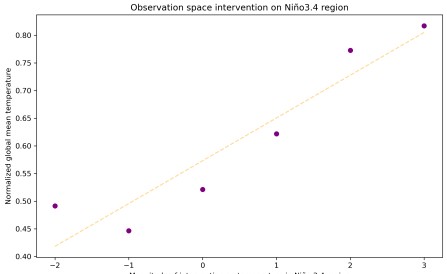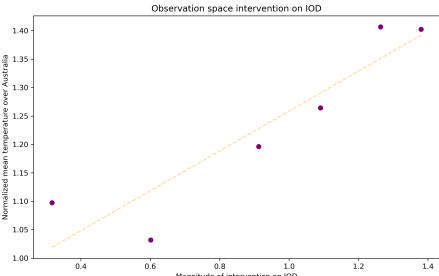

Figure 23: **Intervention in the Niño 3.4 region leads to increased GMST, while intervention in the IOD region increases temperatures in Australia.** (Left) Intervened temperatures for the Niño 3.4 region (x-axis) are positively correlated with GMST (y-axis). (Right) Intervened temperatures for the IOD region (x-axis) are positively correlated with temperatures over Australia (y-axis).

The maps of these interventions are shown in Figure 24, where we intervene to increase the temperature anomaly over Niño 3.4 and Alaska by 2 standard deviations, and we increase the value of the IOD index by 0.5.

## Q  Intervention in latent space

We can also directly intervene on latent variables to explore the effect of interventions on latents which correspond to known modes of climate variability. In Figure 25, which illustrates the effect of intervening on the latent variable that most closely overlaps with the region of the Pacific used to define the Niño 3.4 index.

The single-parent structure allows for a clear correspondence between latents and the physical quantity in the climate model. We verified that the learned Niño 3.4 latent variable is positively correlated with the state of ENSO before performing the counterfactual experiment, by plotting the decoding function from this latent to the corresponding observations (Figure 26). This yielded a linear relationship with a correlation of 0.98.

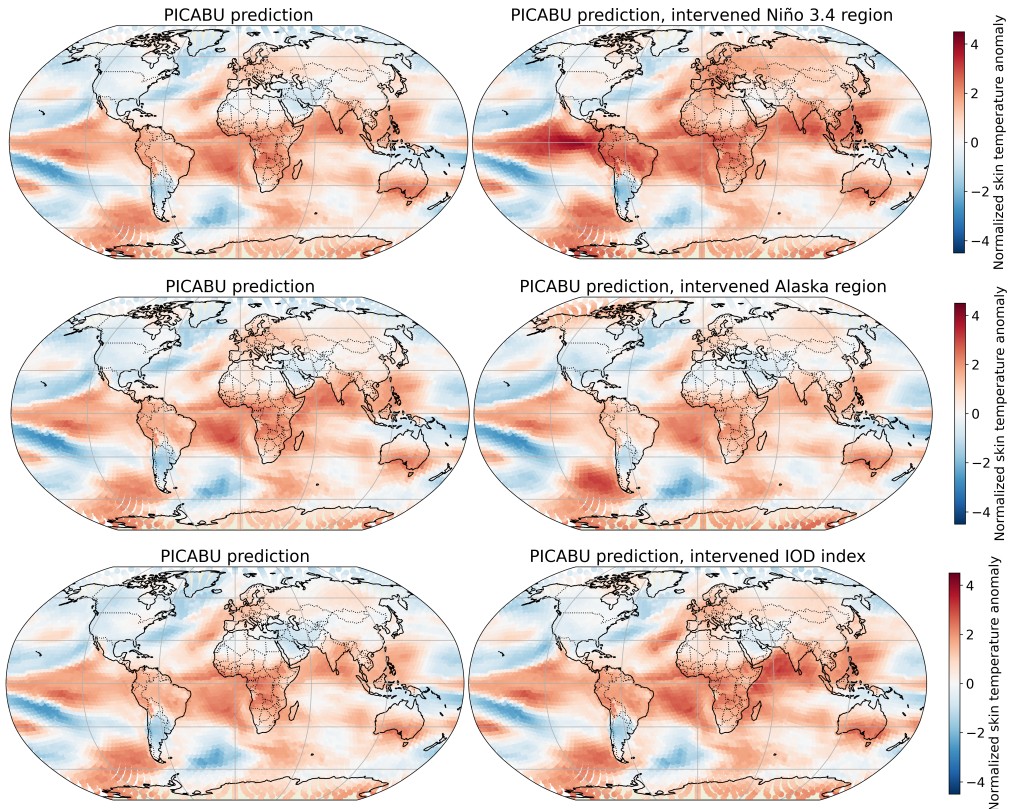

Figure 24: **Maps of the effect of an intervention in observation space where we separately increase the temperature over the Niño 3.4 area and Alaska, and increase the IOD**. (Top row) Increased global temperatures are observed for the Niño intervention, showing a large effect in the tropics and a smaller effect outside the tropics. (Middle row) Little difference is observed when intervening on Alaskan temperatures, despite the similarity of the magnitude of the intervention. (Bottom row) The increase in the IOD index leads to an increase in temperatures over Australia and Africa.

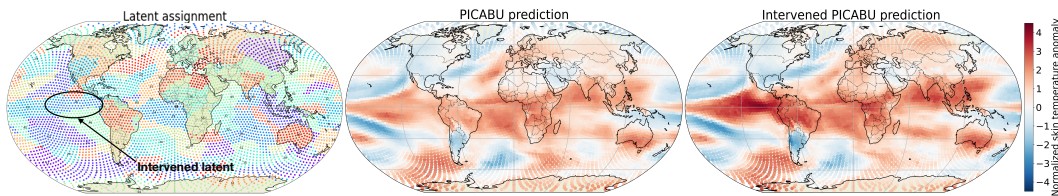

Figure 25: **Intervention on the latent variable that describes the ENSO state.** The left panel shows mapping from latents to observations (colored grid points), and the intervened latent variable. We increase its value, which then influences the latents at the next timestep through the learned causal graph. No other latent variables are intervened on. The middle panel shows the original, unintervened next-step prediction, and the right panel shows the intervened prediction.

# R   Licenses for existing assets

The datasets used in this work are CMIP6, associated with the Creative Commons Attribution 4.0 International license (CC BY 4.0), and CESM2, provided by UCAR. This work builds upon ClimateSet, published in NeurIPS 2023 and publicly available on arXiv, CDSD, publicly available on arXiv, and SAVAR, associated with the GPLv3 license.

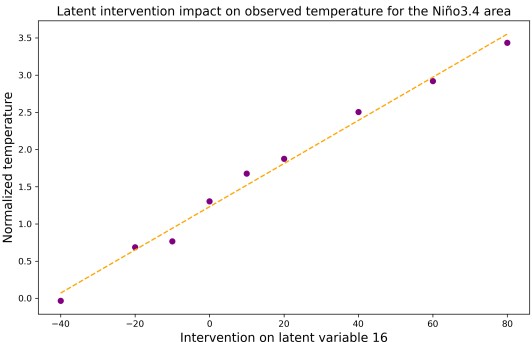

Figure 26: Effect of the intervention on latent variable 16 (most overlapping with the Niño3.4 region) on the physical quantity of temperature in the grid cells within the Niño 3.4 region.

