# OpenReview forum: "Causal Climate Emulation with Bayesian Filtering"
_NeurIPS.cc/2025/Conference — NeurIPS 2025 poster_

### Official Review · Reviewer_fpGY · 2025-06-05

**Clarity:** 2
**Significance:** 3
**Originality:** 4
**Rating:** 4
**Confidence:** 4

**Summary:**

The authors propose a design for a causal emulator of earth system model outputs. The model learns a causal latent representation of the climate model output at past time steps, and use an autoregressive rollout logic combined with Bayesian filtering to emulate climate model output time series. The loss is designed as a composite of multiple components meant to assist the learning of a causal graph in the latent space, and enforce spatiotemporal regularity of the field to stabilize long term predictions. The model is demonstrated using synthetic SAVAR data and NorESM2 simulated surface-temperature fields.

**Questions:**

- Section 3.1: Equation (1) introduces notations for constraints and penalty terms without definition, I couldn't understand how these are different and concretely contribute to the loss?
- Section 3.3 : It is not clear to me why would we want to sample from $p(z^t | x^{\leq t})$. I thought the point of having an autoregressive rollout was that we would be sampling at time $t$ from knowing the past i.e. $p(z^t | x^{<t})$ but I may have missed something.
- Section 4.3 :  If the model does not have any forcing as its input, does it still make sense to compare the next step prediction to the actual next step from the ESM output?
- Table 2 could display the absolute difference to the groundtruth to help readers more easily compare model performances.

**Ethical Concerns:**

["NO or VERY MINOR ethics concerns only"]

**Final Justification:**

I’m still a bit underwhelmed by the evaluation, but the paper offers plenty of good contributions (causal latent structure, spectral regularization, counterfactuals). I’m bumping my score and won’t stand in the way of acceptance.

**Limitations:**

yes

**Quality:**

3

**Strengths And Weaknesses:**

Strengths:
I find the idea of having some form of causal representation within the design of a climate model output emulator very compelling. This is an ambitious and novel endeavor with potentially high impact. In particular, the ability to intervene on a certain factor and trace its spatial impact is very interesting for attribution studies. In that sense, the intervention study presented at the end of the paper is quite compelling.

Weaknesses:
My main concerns with the submission in its current form are two-fold.

First, it is stated at multiple points throughout the manuscript that the model is physics-constrained. However, I find this label to be quite misleading. No explicit physical conservation law is enforced here (e.g. mass, energy). The penalty included under the "physics-constrained" label involve a penalty on the climate model output spectrum, which is of the domain of signal processing, and a constraint on the CRPS which is a scoring rule from decision theory. They certainly do help regularize and stabilize the model, but it is misleading to suggest they encode physics constraints.

Second, I found the experiment section to lack in demonstrating the capabilities for PICABU. The emulation of GMST or Nino34 is a relatively easy target, which could be achieved with simpler time series modeling. I agree it is important to check that PICABU is indeed able to emulate them, but I would expect that demonstrating the capabilities of this model on more challenging field such as precipitation would allow it to shine better. In fact, I would recommend highlighting the intervention study earlier as it's arguably the paper’s most distinctive selling point.

Again, I want to insist that I did find the motivation of the paper very compelling with the manuscript proposing several valuable contributions. There are lots of different components presented in this manuscript, which could make material for more than 1 paper or a longer journal submission, but in its current presentations I found the manuscript perhaps felt a bit rushed.

---

> ### Author Rebuttal · Authors · 2025-07-30
>
> We thank the reviewer for noting the very compelling motivation as well as several valuable contributions of our paper.
>
> We would like to first **answer the weaknesses raised by the reviewer** point-by-point:
>
> **Physics-informed losses**
>
> First, we would like to clarify that we do not use the term “physics-constrained” in the paper, but rather “physics-informed” as the spectral losses and the Bayesian filter use the spatial spectrum of the data to inform or correct the prediction. However, we agree with the reviewer that this term is a little strong for the usage of the losses and we will change the Section 3.2 title from “Physics-informed loss functions” to “Additional loss terms” since the CRPS loss is indeed not related to physics.
>
> **Demonstrating the capabilities of PICABU**
>
> Second, we want to clarify that we build a dynamical emulator that simulates whole fields at monthly timesteps; we are not solely building an emulator of ENSO or of GMST. More precisely, PICABU takes as input only 5 consecutive months of the whole field of surface temperature, and predicts 1200 months (100 years) for the whole temperature field. We then evaluate its performance by looking at statistics (Mean, Std. Dev., Range, LSD) and Power Spectral Density (PSD) of GMST, ENSO (Table 2, Fig. 3), as well as the Indian Ocean Dipole (IOD) and the Atlantic Multidecadal Oscillation in Appendix H (Fig. 14, Table 4). The statistics inform us about the bias and variance of the emulated modes, and the PSD informs us if the long-term dynamics of these modes are well reconstructed. Even though the model learns a graph describing the latent dynamics happening during only 5 months, and autoregressively predicts each subsequent month given its previous 5 months, we observe correctly reconstructed long-term properties of the climate model data, such as the ENSO 3-year oscillation. This indicates that **the underlying dynamics learned by the model are accurate for main modes of variability**.
>
> To illustrate that this is a hard task, we want to note that **the two models we compared to (an extension of V-PCMCI to linearly predict the next step + a Vision transformer) fail to accurately capture the dynamics of the 4 modes considered**. In general, **producing long-term stable and physical rollouts with machine learning models is very challenging**, due to error growth from various sources (Chattopadhyay et al., 2024; Karlbauer et al., 2024). The ablated PICABU models also perform worse at simulating the dynamics, showing the difficulty of the task: without CRPS to help the model capture the range of temperature distributions or the spectral terms of the loss to alleviate issues with the MSE loss such as the double-penalty of MSE, PICABU does not perform as well.
>
> We agree with the reviewer that simulating precipitation fields is more challenging and would better highlight the capacities of the model. In this manuscript we focused on emulating a single field, temperature, and extending to further variables is certainly planned future work, as mentioned in the Discussion section.
>
> We thank the reviewer for highlighting the counterfactual study. This is indeed one of the paper's selling points, and one of the main reasons for using a constrained model, rather than an alternative, more flexible model: the identifiability guarantees allow us to explore the learned model by doing counterfactual experiments. To further strengthen this aspect of this paper and illustrate possibilities for attribution studies with PICABU, we have carried out further counterfactual experiments (e.g. analysing the effect of perturbing the IOD on Australian temperatures), which are detailed in the response to Reviewer 5AXT under **Counterfactual experiments**. The single-parent structure and sparsity constraint make it more challenging to accurately perform next-step prediction, which is an additional reason why it is impressive that the model is able to accurately simulate long-term dynamics of the main modes of climate variability.
>
> We would like to now **answer the reviewer's questions**:
>
> - Thank you for raising this point. The details of all terms in equation 1 can be found in Appendix B, but we will clarify the notation and add more details in the main text to clarify this and make the reading easier.
> - We indeed sample from $p(z^t \mid x^{\textless t})$ for autoregressive rollouts; we apologize for not making this more clear. Equation 5 was meant to illustrate the classical Bayesian filter framework, which is not designed for autoregressive rollouts but for estimating latent states from observations. We will reformulate the text to make this clearer.
> - This is a great point. We agree that comparing the prediction to the ESM output at long lead times does not make sense, but, at short lead times (e.g. 1 timestep / next step prediction), we believe it represents an interesting test of how our emulator can generalise to initial conditions outside its training distribution. The change in forcing is small for one timestep (one month), and we are thus testing the model on a physically plausible but out-of-distribution set of initial conditions. Naturally, we see considerable degradation in performance, which is tied to the lack of external radiative forcing as input (and in the training data), but it is interesting that the model with causal components generalises better than the models without these components.
> - This is a good idea, and we will add this to the final manuscript.
>
> **We thank the reviewer again for the insightful comments and hope that the reviewer’s concerns are addressed by our responses.**

---

> > ### Comment · Reviewer_fpGY · 2025-08-04
> > **Response to authors**
> >
> > Sorry for being late to engage in the discussion period, I was mostly offline last week. Thanks a lot for the detailed response and for addressing in depth my points and questions.
> >
> > - Physics-informed wording:  My mistake you did write physics-informed, not physics-constrained, I was a bit too fast with this. That said, even physics-informed still feels a bit strong given there’s no explicit conservation law. I like the softer language proposed by the authors.
> >
> > - PICABU Evaluation : I thank the authors for their additional explanations on the performances of PICABU. I appreciate that autoregressive climate emulation is indeed a hard task and that it's challenging to keep it stable, something that PICABU seems capable of doing. Maybe I just lack familiarity with how things are typically evaluated for autoregressive models, but the evaluation still feels a bit lacking to me. What I mean is that I could probably fit an ARIMA model for each time serie considered here (GMST, ENSO, IOD, AMO). Evaluating PICABU on its ability to reproduce the dynamics of these modes is an important sanity check for sure, and it's great to see it's working. But it doesn't tell me much about the produced spatial fields for example.
> >
> > Overall, while I’m still a bit underwhelmed by the evaluation, the paper offers plenty of good contributions (causal latent structure, spectral regularization, counterfactuals). I’m bumping my score and won’t stand in the way of acceptance.

---

> ### Author Response · Authors · 2025-08-06
>
> We thank the reviewer for noting the multiple good contributions and for increasing their score.
>
> The reviewer raises a very interesting point on the evaluation of autoregressive models. It is indeed instructive to compare our model output to ARIMA models fitted on each separate mode (GMST, ENSO, IOD, AMO). Such models have already been developed (for GMST: Nooteboom et al., 2018; Yu et al., 2023; for Global Mean Sea-Level: Elneel et al., 2024;  for regional temperatures: Lai et al., 2020; Rosmiati et al., 2021). Authors point out that ARIMA methods lead to accurate short-term predictions but poor long-term predictions, and need to be extended to capture the non-linear evolution of these time-series (Nooteboom et al., 2018).
>
> An extension of these models are Bayesian Vector Autoregressive models, which to the best of our knowledge haven't been applied to model full fields of climate variables but have shown good results for studying climate sensitivity (Goodwin et al., 2024).
>
> We will discuss these points in the revision. Thank you very much for this suggestion.

---

### Official Review · Reviewer_k59w · 2025-07-01

**Clarity:** 4
**Significance:** 3
**Originality:** 3
**Rating:** 4
**Confidence:** 5

**Summary:**

This paper proposes PICABU, a physics-informed causal emulator for climate‐model output. Building on the CDSD latent-causal discovery backbone, the authors (i) add spectral and CRPS losses that encourage the emulator to respect spatial- and temporal-frequency statistics, (ii) introduce an explicit Bayesian particle–filter to stabilize very long autoregressive roll-outs, and (iii) demonstrate basic counterfactual experiments by intervening on learned latents. On the synthetic SAVAR benchmark, PICABU recovers ground-truth graphs with near-perfect F1, and on 800 years of NorESM2 data, the model produces century-scale roll-outs whose global-mean temperature and ENSO spectra track the reference more closely than a ViT baseline or V-PCMCI.

**Questions:**

I think this is a good paper and it should be accepted. But, the lack of significant results for PICABU on top of CDSD are concerning w.r.t. novelty. I would like to see:
1. The evaluation criteria and inference test results clearly stated.
2. A more detailed discussion on physics-informed losses and why these losses have anything to do with the specifics of this data.
3. Results on real data for the counterfactual and causal graph analysis.

**Ethical Concerns:**

["NO or VERY MINOR ethics concerns only"]

**Final Justification:**

The authors provided a comprehensive set of responses to my questions. They provided significance results as well as clarifications on all of the weaknesses I highlighted.

**Limitations:**

Yes

**Quality:**

2

**Strengths And Weaknesses:**

# Strengths

1.  PICABU integrates causal discovery with explicit physics-oriented regualizers (spectral matching + CRPS) and a sparsity-constrained DAG, yielding an interpretable latent space
2. The Bayesian filter prevents error blow-up and is described rigorously. 100-year NorESM2 roll-outs remain unbiased and capture key ENSO peaks.
3. Section 4.4 shows the counterfactual benefits of PICABU by demonstrating El Niño intervention. Figure 4 is well presented!
4. Ablations are present in this work, something which comparable works frequently lack.
5. The writing is lucid and well-structured. The paper was easy to read.

# Weaknesses
1. L128: CRPS is a generic probabilistic score. The authors never justify why it is "physics-informed" or what physical invariants it helps enforce. CRPS as an enforcement of the autoregressive nature of the underlying phenomenon is not specific to climate emulation.
2. I would have liked to see some discussion or demonstration of PICABU on real-world data. Causal accuracy is validated exclusively on SAVAR. See something like Debeire et. al. 2024. [0]
3. L192-194: SAVAR link strengths are drawn from Beta(4, 8) with $\mu = 1/3$ and $\sigma = 0.13$, but these are essentially magic numbers. Are these defaults? Do they come from another paper (if so, cite), or were they chosen through tuning?
4. L206: Table 1 shows that CDSD already performs at 0.98-1.00 F1; without error bars it is unclear that PICABU's gains are significant. Further, it looks like all results in Table 1 are already highly saturated? The underlying contribution is hard to fairly assess given this lack of reporting.
5. L208-209: The authors point out that SAVAR datasets are low complexity and therefore do not require the physics-informed losses added by PICABU. So, why use this to demonstrate your work? It really takes away from the novelty.
6. While ablations are provided (partially in Table 2),  the authors never show a run with all physics losses removed, nor a plain MSE baseline, so the marginal value of the spectral + CRPS loss terms are hard to judge.
7. The evaluations throughout the paper are under-specified. The F1 score in table does not explain how true positives/negatives are identified for weighted graphs. Similarly, Table 3 reports statistical significance via a t-test but omits exact p-values and the choice of $\alpha$. If a statistical test is going to be conducted, then I'd like to see these values reported at least in the Appendix.
8. None of the results for PICABU in Table 3 are statistically significant over their removed constraints. I'm not sure what the constraints are then adding?
9. Figure 1 is a bit hard to read. The "autoregressive rollouts" arrow is hidden, with all the red, and the "loss function" half ellipse is hard to see.
10. L262/L264: Figures 9 and 11 are referenced, but they do not exist in the main paper. They exist in Appendix F and should be referenced as such.
10. L820-822: The appendix mentions "This work builds upon ClimateSet, published in NeurIPS 2023 and publicly available on arXiv, CDSD, publicly available on arXiv, and SAVAR, associated with the CC BY 4.0 license." I'm worried that this breaks double blind review? (hello Rolnick lab). Also, isn't SAVAR GPLv3, not CC BY 4.0? [1]


[0] Debeire, K., Bock, L., Nowack, P., Runge, J., and Eyring, V.: Constraining uncertainty in projected precipitation over land with causal discovery, EGUsphere [preprint], https://doi.org/10.5194/egusphere-2024-2656, 2024.

[1] https://github.com/xtibau/savar/tree/master

---

> ### Author Rebuttal · Authors · 2025-07-30
>
> We thank the reviewer for writing that the paper is good and should be accepted, for finding it well-structured and easy to read, and for noting the rigorous description of the Bayesian filter and ablation studies.
>
> We would like to address the reviewer's questions, first by clarifying the **novelty w.r.t. CDSD**, and the importance of the additional components.
>
> Without the novel Bayesian filter, the autoregressive predictions of all models diverge. Moreover, even though SAVAR experiments and the results reported in Table 2 do not show significant differences between CDSD, PICABU, and its ablated models (No spectral / No CRPS), Appendix F Fig. 10 clearly shows that **without CRPS or the spectral term in the loss, the model does not capture the ENSO 3-year oscillation and has too much power for low frequencies, showing that those terms are key to capturing accurate climate dynamics**. The CRPS term improves the range of temperatures and reduces smoothing (Lang et al., 2024), while the spectral terms encourage better representation of spatiotemporal characteristics and maintain physical realism (Kochkov et al., 2024; Chattopadhyay et al., 2024). We will include in the revised manuscript a detailed discussion of these terms and their contributions to the specifics of the emulation task.
>
> As requested by the reviewer, we now clearly state **the evaluation criteria and inference test results**. At inference, PICABU takes as input only 5 consecutive months of the whole field of surface temperature, and predicts 1200 months (100 years) for the whole temperature field. We then evaluate its performance by looking at statistics (Mean, Std. Dev., Range, LSD) and Power Spectral Density (PSD) of GMST, ENSO (Table 2, Fig. 3), as well as the Indian Ocean Dipole (IOD) and the Atlantic Multidecadal Oscillation in Appendix H (Fig. 14, Table 4). The statistics inform us about the bias and variance of the emulated modes, and the PSD informs us if the long-term dynamics of these modes are well reconstructed. Even though the model learns a graph describing the latent dynamics happening during only 5 months, and autoregressively predicts each month given its previous 5 months, we observe correctly reconstructed long-term properties of the climate model data, such as the ENSO 3-year oscillation. This indicates that **the underlying dynamics learned by the model are accurate for main modes of variability**.
>
> **Causal graph analysis**
>
> We do not seek to validate the learned causal graphs directly, as we do not have access to true causal graphs for real–world or climate model data. We evaluate our model by emulating climate models (which aim to simulate the real-world climate system) and **showing that the learned causal graph leads to accurately predicted long-term dynamics, and physically-reasonable responses to perturbations**. In Debeire et al., the goal instead is climate model evaluation, which is performed by comparing causal graphs learned from climate model data to causal graphs learned from real-world Earth system observations or reanalyses. While PICABU could be used for this, it is a different research application to climate model emulation, the primary goal of our paper.
>
> However, to strengthen the analysis of the causal graph learned by PICABU, **we completed a set of additional counterfactual experiments to compare the causal relationships learned by PICABU to well-established causal relationships in climate models** (unfortunately, images cannot be included):
> * We extended our ENSO counterfactual experiment by intervening and decreasing/increasing the temperature in the Niño 3.4 index region between -3 and 3 degrees, and found, in line with established physical knowledge, that these interventions led to decreased/increased GMST with a correlation of 0.9.
> * We carried out counterfactuals in the region over Alaska, finding that similar interventions in this region had very little effect on global temperatures (correlation = 0.08). This aligns with the physical knowledge that perturbations in polar regions have much more localised effects than perturbations in the tropics.
> * We intervened on the IOD, and found that the IOD-intervened temperatures are highly correlated with Australian temperatures (correlation = 0.87). This is a well-known effect (e.g. Cai et al., 2009) accurately modelled by PICABU.
>
> We believe these experiments provide much stronger evidence for the model's ability to learn physically reasonable causal dynamics, well-aligned with known physical teleconnections, and improve our causal graph analysis. We will describe them clearly in the updated manuscript.
>
> **We now would like to address the weaknesses outlined by the reviewer.**
>
> 1. We agree with the reviewer and will change the title of Section 3.2 from “Physics-informed loss functions” to “Additional loss terms”, since CRPS is indeed not physics-informed. Thank you for raising this, we hope the terminology is now more appropriate.
> 2. It is difficult to validate causal accuracy on real-world data, as the “true” causal graph is unknown. Debeire et al. instead seek to validate climate models against observational data. They compare the graph estimated by Varimax-PCMCI (which fails to recover the causal graph in the SAVAR data) when run on observation data vs. climate model data, not claiming that any of the graphs are the “true” causal graph. In general, however, it is absolutely right that further validation of these methods on real-world data would be helpful, and exploring new benchmark datasets as well as evaluating skill and generalization on real-world data is planned future work.
> 3. These numbers were chosen because Beta(4, 8) is a distribution in [0,1] with low probability around 0 and 1, to avoid autoregressive weights that are too low or too high (the mean and variance fall out of the choice of the parameters of the Beta distribution, which were picked as “nice round numbers”). We agree that this choice seems a bit arbitrary but it was done when generating the data before running the models, and not to tune the results. We will add results with more distributions in the updated manuscript.
> 4. This is a fair concern. Table 1 does not show clear gains of PICABU vs CDSD. As mentioned above in the response, such gains are shown for the climate model data in Appendix F, Fig. 10. We initially separated Fig. 10 from Fig. 3 for clarity, but will put it back in the main text to better highlight the novelty and the importance of our additions to CDSD.
> 5. Unfortunately, there are not many climate-relevant simulated datasets that can be used to evaluate causal representation learning methods and in that regard we view SAVAR as state-of-the-art and appropriate for evaluating our method. It is important to first validate that our method recovers the SAVAR causal graph before applying it to climate model data. Moreover, Varimax-PCMCI, the method used by Debeire et al., and widely adopted in climate science, fails on SAVAR, showing that it is not a simple task. Our additions to CDSD are key to getting good results on climate data but do not lead to significant differences on SAVAR.
> 6. We haven’t reported a plain MSE baseline in Table 2 as it diverges even with the Bayesian filter. We show that the model diverges when only removing the sparsity constraint and will add a line showing that a plain MSE baseline diverges. Similarly, we show results without CRPS and without the spectral terms in Table 3 and Appendix F, Fig. 10, where we see the necessity of these additional losses.
> 7. We show here the False Positives/Negatives corresponding to Table 1. As PICABU implements sparsity as a constraint rather than a penalty, we are comparing all three models when they retain the correct total number of expected links. Hence, the number of false positives and false negatives are generally equal. We are reporting “TP,FN” here:
>
>  | N latents|4| 4|4|4|25|25|25|25|100|100|100|100|
> |-|-|-|-|-|-|-|-|-|-|-|-|-|
> | Difficulty | Easy | Med-easy | Med-hard | Hard | Easy | Med-easy | Med-hard | Hard | Easy | Med-easy | Med-hard | Hard |
> | N links| 4| 8| 13| 11| 25| 49| 75| 324|100| 200| 300| 5049 |
> | PICABU| 0,0| 0,0| 0,0| 0,0| 0,0| 2,1| 4,4| 60,60| 2,2| 2,2| 14,14| 796,796|
> | CDSD| 0,0| 0,0| 0,0| 2,2| 1,1| 0,0| 3,3| 56,56| 2,2| 4,4| 12,12| 1021,1021 |
> | V-PCMCI| 0,0| 1,1| 6,7| 3,4| 6,6| 28,27| 43,43| 252,252  | -| -| -| -    |
>
> We show here the p-values corresponding to the significance tests performed in Table 3. We chose $\alpha = 0.05$ for the significance tests. We apologize for not reporting it and will add it to our appendix.
>
> | Scenario| | picontrol | | | SSP2-4.5 | | | SSP3-7.0 | || SSP5-8.5 ||
> |-|-|-|-|-|-|-|-|-|-|-|-|-|
> | Metric| MAE | R²| LSD | MAE | R²| LSD | MAE | R²| LSD | MAE | R²| LSD |
> | No sparsity |1.9e-4| 0.012 |1.5e-8| 6.7e-5|0.00046| 4.0e-5| 0.047 |0.042|0.85|5.3e-3 |0.021| 0.036 |
> | No ortho|  9.5e-8|1.03-4|0.049|3.7e-6|4.3e-6|0.022| 0.021| 2.3e-3|0.76|7.9e-5| 6.3e-5| 0.047|
>
> 8. The PICABU results in Table 3 are indeed significant compared to PICABU with removed constraints. The ‘*’ in the “No sparsity” and “No ortho” rows indicates statistical significance compared to PICABU. We are sorry this was not presented clearly - we will clarify this in the text and table.
> 9. We will fix the figure; thank you for noting this.
> 10. We will ensure that the appendix figures are referenced as such.
> 11. The NeurIPS checklist asks to ensure that “the license and terms of use [are] explicitly mentioned”. As we build upon CDSD, and use SAVAR and CMIP6 datasets, we added this section in the appendix. This information is available online and we’re only reporting it. You are correct that SAVAR is GPLv3 and we apologize for this error. We will correct it.
>
>
> **We thank the reviewer for the helpful comments and questions. We hope that our responses address the reviewer’s concerns and will integrate all points addressed above into the revised manuscript.**

---

> > ### Comment · Reviewer_k59w · 2025-08-05
> >
> > Thank you, these additions address my concerns. I will re-consider my score accordingly.

---

> ### Author Response · Authors · 2025-08-06
>
> Thank you for acknowledging that it addresses your concerns and reconsidering your score.
>
> We would like to thank you again for the thoughtful review, and we will update the manuscript with the different points described in the rebuttal as we believe your review helps improve the quality of our paper.

---

### Official Review · Reviewer_yBUD · 2025-07-01

**Clarity:** 3
**Significance:** 3
**Originality:** 3
**Rating:** 4
**Confidence:** 5

**Summary:**

This paper introduces PICABU, a novel framework for climate emulation that aims to solve the core challenges of long-term instability and lack of physical interpretability in traditional data-driven models. Building upon a causal representation learning framework (CDSD), the authors make two main contributions: (1) integrating physics-informed loss functions, particularly spatial and temporal spectral losses, to enforce physical consistency; and (2) a novel Bayesian filtering mechanism that uses the data's spatial power spectrum as a physical anchor to ensure stable, long-term autoregressive rollouts. The experiments demonstrate that PICABU not only accurately recovers known causal relationships and performs stable 100-year climate emulations, but its causal structure also enables interpretable counterfactual experiments, a key feature for scientific analysis.

**Questions:**

See Weekness

**Ethical Concerns:**

["NO or VERY MINOR ethics concerns only"]

**Limitations:**

See Weekness

**Quality:**

3

**Strengths And Weaknesses:**

Strengths

Exceptional Novelty in Ensuring Stability: The paper's most significant contribution is the proposed Bayesian filtering strategy. Using a physically-invariant quantity like the spatial spectrum as a proxy observation to correct state drift during autoregressive rollouts is a highly innovative and effective idea. It provides a powerful solution to the critical problem of long-term instability in ML emulators.

Strong Physical Grounding and Interpretability: By incorporating physics-informed losses and preserving a causal structure, PICABU moves beyond being a "black-box" emulator. The model not only produces outputs that are statistically consistent with climate data but also provides an interpretable causal graph. The counterfactual experiment is a compelling showcase of this capability, highlighting its potential for scientific attribution studies, which is a crucial advantage over many other deep learning models.

Comprehensive and Rigorous Experimental Validation: The paper's evaluation is thorough and convincing. It first validates the model's ability to recover causal graphs on synthetic data (SAVAR) and then systematically demonstrates its long-term emulation capabilities on two real-world climate models (NorESM2 and CESM2-FV2). The extensive ablation studies clearly prove the necessity and effectiveness of each of the paper's innovations (spectral losses, the Bayesian filter, and the causal constraints).

Methodological Robustness: The model demonstrates similarly strong performance on two different climate models with only minimal hyperparameter tuning, suggesting that the approach is robust and generalizable.

Weaknesses

Limitations of the Single-Parent Assumption: As the authors correctly identify in the discussion, the single-parent decoding assumption is a primary limitation. This constraint partitions the observation space (the globe) into non-overlapping regions, each dominated by a single latent variable. This may struggle to fully capture the complex, overlapping influences found in the real climate system and could be a reason for the observed high-frequency noise when emulating large-scale, smooth variables like the Global Mean Surface Temperature (GMST).

Computational Overhead of the Filter: While vastly faster than running a full Earth System Model, the Bayesian filtering step requires N*R sampling, decoding, and FFT operations at each timestep, adding computational cost compared to a simple deterministic rollout. A brief analysis of this overhead would be a useful addition.

---

> ### Author Rebuttal · Authors · 2025-07-30
>
> We thank the reviewer for highlighting the novelty of the Bayesian filtering strategy, the physical grounding, interpretability, and rigorous validation of our model, as well as the robustness of our method.
>
> We would like to answer the weaknesses raised by the reviewer:
>
> **Single-parent assumption**
>
> The single-parent assumption is indeed a strong assumption but is not so limiting as to prevent the model from being a useful tool for climate scientists. The assumption imposes that the latents correspond to single climate modes in a geographic region, that ‘couple’ or interact through the causal graph. **The learned latents thus closely mirror simplified climate indices (e.g. scalar indices representing climate variables in fixed regions of space such as ENSO, NAO, MJO, IOD) that climate scientists widely rely on to describe large-scale atmospheric dynamics and teleconnections.** Climate scientists often use simple representations of complex processes for interpretable, parsimonious description of large-scale phenomena; our work parallels such methods closely.
>
> Furthermore, we argue that the single-parent assumption is a necessary and principled first step towards interpretable and trustworthy data-driven climate models. By enforcing the single-parent structure, we gain theoretical guarantees of causal identifiability, allowing for the robust counterfactual experiments that are a primary contribution of this work. **This assumption makes our tool more interpretable and easier to visualize for climate scientists, and allows exploring the effect of counterfactual experiments, which we hope will be useful for attribution studies by domain scientists**. This is a necessary alternative to more flexible emulator models, which can be more powerful in terms of predictions but are much less interpretable.
>
> To clarify this for the reader we will add a dedicated paragraph to the Discussion section to explicitly detail the types of physical interactions and modelling this assumption is appropriate for.
>
> **Computational requirements of the Bayesian filter**
>
> The Bayesian filter is more expensive than a deterministic filter, but operations (sampling, decoding and FFTs) can be batched at each step and performed rapidly. **Typically, we are able to simulate 100 years (1200 months) in less than 6 minutes on a single RTX8000 GPU with 48GB memory, with $N=300$ and $R=10$ (with the computational time scaling linearly with $N, R$)**. This is a very low computational requirement, even at inference time, compared to other models. Upon acceptance, we will add a formal analysis of the computational cost of the Bayesian filter as well as training of our model in our appendix; thank you for suggesting this.
>
> **We thank the reviewer again for the positive review and insightful comments and hope that the reviewer’s concerns are addressed by our responses.**

---

### Official Review · Reviewer_5AXT · 2025-07-02

**Clarity:** 3
**Significance:** 3
**Originality:** 3
**Rating:** 3
**Confidence:** 2

**Summary:**

This paper introduces PICABU, a novel climate model emulator designed for fast, stable, and interpretable long-term simulations. It integrates causal representation learning to identify key climate drivers, physics-informed losses to ensure physical consistency, and a Bayesian filter to maintain stability during autoregressive rollouts. Experiments on synthetic and real-world climate data show that PICABU accurately recovers causal structures and captures major climate variability like ENSO.

**Questions:**

See weakness

**Ethical Concerns:**

["NO or VERY MINOR ethics concerns only"]

**Final Justification:**

In the rebuttal, the authors comprehensively supplement experiments on counterfactual reasoning and the variance of the Bayesian filter. With the additional clarifications on how their method numerically manipulates the latent variables, its viability is more substantiated. On this end, my concerns are well addressed.

My only remaining concern is that incorporating these revisions might result in substantial changes to the original submission. I sincerely hope the AC considers these factors.

**Limitations:**

The authors have already acknowledged the limitations of their work (i.e., reliance on the single-parent decoding assumption). However, I still somehow find it unreliable for a global climate modeling system.

**Quality:**

2

**Strengths And Weaknesses:**

## Strengths
- The proposed PICABU model integrates causal representation learning with physics-informed spectral losses and a novel Bayesian filter, presenting a well-motivated and cohesive approach to climate emulation.
- The paper's claims are supported by a comprehensive suite of experiments, including causal graph recovery (Table 1) and ablation study (Table 2).
- The model shows strong generalization performance to out-of-distribution climate scenarios (Table 3).

## Weaknesses
- The authors state that model performance depends on "considerable manual tuning" of hyperparameters. PICABU seems to be dependent on quite a lot of configurable parameters as shown in Table 6. These raise concerns regarding the applicability and practical adoption of PICABU.
- The justification for some modeling choices could be stronger; for instance, the variance $\tilde{\sigma}$ in the Bayesian filter is simply "assumed constant through time". The authors may provide a further analysis of this simplification's impact.
- While the authors acknowledge that the main limitation of their model arises from the single-parent decoding assumption, I still find it somewhat difficult to accept that a system built upon a premise fundamentally at odds with real-world physics can be considered fully sound. The assumption of non-overlapping, single-factor influence contradicts the highly coupled nature of the climate system.
- I find the counterfactual experiment in Section 4.4 difficult to interpret as strong evidence for causal discovery. First, the experiment appears to cherry-pick the easiest case; demonstrating that a stronger ENSO leads to higher global mean temperatures is not a particularly surprising finding. Second, the physical realism of the intervention is questionable. What does "manually setting the latent variable to a new, higher value" correspond to physically? Does it represent a 10% reduction in trade winds or an increase in subsurface warm water upwelling? Is it possible that the learned latent variable is, in fact, negatively correlated with the state of ENSO, in which case the interpretation of the intervention would be reversed?

---

> ### Author Rebuttal · Authors · 2025-07-30
>
> We thank the reviewer for noting the well-motivated and cohesive approach to climate emulation, the novelty of the Bayesian filter, and the comprehensive suite of experiments.
>
> We would like to **answer the weaknesses raised by the reviewer** point-by-point:
>
> **Hyperparameter tuning**
>
> Thank you for noting this important point. We agree that the current description of “considerable manual tuning” is vague and would like to clarify it further.
>
> First, we want to note that the “considerable manual tuning” was only done upfront in order to establish a robust set of default values. More precisely, we performed a search over the ~20 parameters described in Appendix N Table 6, with ~100 runs of PICABU. Although described as “considerable manual tuning”, this search remains reasonable compared to standard hyperparameter searches for ML models.
>
> After this initial set of parameters was selected on NorESM, we only performed minimal parameter tuning to train PICABU on a second climate model (CESM2), as indicated in Appendix I. The initial hyperparameters transferred directly and effectively to this different climate model, though performance was further improved by a single hyperparameter adjustment, and performed well on the simulated dataset (SAVAR) where only the sparsity of the final graph was changed. **These results suggest that in practice, future users of the model will not have to complete considerable tuning when using our current hyperparameters as a baseline**. Moreover, Appendix L shows that our default hyperparameters are robust while varying other aspects of the model (e.g. number of latents, sparsity).
>
> We will describe these findings in detail in the Appendix as we agree this is very important for potential users, and the current “considerable manual tuning” description is vague - thank you for raising this issue.
>
> **Variance of the Bayesian filter**
>
> We agree with the reviewer that the choice of constant variance is not well justified in the original text. It is indeed more standard to estimate the variance when running a Bayesian filter and we thank the reviewer for suggesting this update to our Bayesian filter algorithm. We ran experiments using the model’s estimated variance and report the results in the following table.
>
> |        |       |  GMST    |       |         |    |   Niño3.4     |        |   |   |    IOD   |      |  | | AMO | | |
> |-|-|-|-|-|-|-|-|-|-|-|-|-|-|-|-|-|
> | |Mean|Std Dev.|Range|LSD|Mean|Std Dev.|Range|LSD|Mean|Std Dev.|Range|LSD|Mean|Std Dev.|Range|LSD|
> |**Ground truth**| 0 | 0.177 | 1.357 | 0 | 0 | 0.927 | 5.73 | 0 | 0 | 0.877 | 7.36 | 0 | 0 | 0.368 | 2.58 | 0 |
> |**PICABU**|-0.0518|0.277|2.02|0.444|-0.0914|1.05|5.99|0.191|0.186|0.753|5.36|0.0772|0.0538|0.445|3.27|0.248|
> |**PICABU est. var.**|-0.0742|0.324|2.15|0.464|-0.198|1.27|7.70|0.206|0.110|0.845|5.46|0.0753|0.0157|0.498|3.27|0.261|
>
> We unfortunately cannot upload figures to illustrate the updated results but when using the estimated variance, the statistics for ENSO, GMST, and AMO are very slightly degraded, because for these quantities the variance of the model is slightly higher than the ground truth’s (Table 2), and this gets propagated throughout the prediction when using the estimated variance. Given this empirical finding that the constant variance term helps to constrain the model for these quantities, we will maintain the results from our current implementation, and describe this analysis in detail in the manuscript.
>
> **Single-parent assumption**
>
> The single-parent assumption is indeed a strong assumption but is not so limiting as to prevent the model from being a useful tool for climate scientists. The assumption imposes that the latents correspond to single climate modes in a geographic region, that ‘couple’ or interact through the causal graph. **The learned latents thus closely mirror simplified climate indices (e.g. scalar indices representing climate variables in fixed regions such as ENSO, NAO, IOD) that climate scientists widely rely on to describe large-scale atmospheric dynamics and teleconnections**. Climate scientists often use simple representations of complex processes for interpretable, parsimonious description of large-scale phenomena; our work parallels such methods closely.
>
> Furthermore, we argue that the single-parent assumption is a necessary and principled first step towards interpretable and trustworthy data-driven climate models. By enforcing the single-parent structure, we gain theoretical guarantees of causal identifiability, allowing for the robust counterfactual experiments that are a primary contribution of this work. **This assumption makes our tool more interpretable and easier to visualize for climate scientists, and allows exploring the effect of counterfactual experiments, which we hope will be useful for attribution studies by domain scientists**. This is a necessary alternative to more flexible emulator models, which can be more powerful in terms of predictions but are much less interpretable.
> To clarify this for the reader we will add a dedicated paragraph to the Discussion section to explicitly detail the types of physical interactions and modelling this assumption is appropriate for.
>
> **Counterfactual experiments**
>
> The counterfactual experiment shown in section 4.4 is not presented as strong evidence of causal discovery, but is rather an example of the exploration that the model allows. Thanks to the identifiability guarantees that the single-parent assumption ensures, this model enables proper counterfactual experiments, which we illustrate in Section 4.4.
>
> ENSO is a well-established climate phenomenon, with well-studied causal links, and we therefore consider it the first go-to experiment in climate science when it comes to counterfactuals. We understand that it is not surprising that a stronger ENSO leads to higher global mean temperatures, but we consider it important to verify that the model performs as expected for this canonical case.
>
> We also understand the concerns about the physical realism of the intervention. We agree that in the paper, “manually setting the latent variable to a new, higher value” is indeed unclear. However, the single-parent structure allows for a clear correspondence between our latents and a physical quantity. Your last point is absolutely correct, and we did verify that the learned variable is positively correlated with the state of ENSO before performing the counterfactual experiment, by plotting the decoding function from this latent to the corresponding observations. This yielded a relatively linear relationship with a correlation of 0.93. In the shown experiment, the intervention corresponded to an average increase of the ENSO temperatures of 0.25 degrees. We will add these details in the manuscript, and will plot the decoding function mapping the latents we intervene on to the observations, showing what the intervened latent precisely corresponds to in observation space.
>
> It is a reasonable point that our initial counterfactual experiment was somewhat limited. **We have now run a comprehensive set of new experiments that we believe provide additional strong evidence for the model's ability to learn physically reasonable causal dynamics**. These experiments, described below, will be detailed in the main text and illustrated in the paper, to highlight the more nuanced teleconnections also learned by PICABU.
>
> First, we reversed the direction of the ENSO counterfactual, and as expected found that reducing the strength of ENSO led to lower global temperatures (GMST). We carried out experiments for a number of different latent values to explore the effect of different interventions. We found a strong positive correlation between the latent intervention and GMST overall (correlation = 0.87).
>
> Moreover, the identifiability guarantees of the model allow us to do the intervention in the observation space rather than in the latents – i.e. we can directly modify the temperatures in a given region, rather than modifying the latent values, for better interpretability of the intervention. **We carried out a number of experiments with interventions in observation space to illustrate this and further check that the learned causal relationships align with additional well-studied physical teleconnections**. Namely, we completed the following analyses and will add them to our paper (figures are not permitted in the rebuttal):
>
> * We intervened in the region of the Niño 3.4 index to decrease/increase the temperature in this region between -3 and 3 degrees, and found, in line with established physical knowledge, that these interventions led to decreased/increased GMST with a correlation of 0.9.
>
> * We carried out counterfactuals in the region over Alaska, finding that identical interventions there had very little effect on GMST (correlation = 0.08). This finding aligns with general principles of atmospheric dynamics that perturbations in polar regions have much more localised effects than perturbations in the tropics.
>
> * We intervened on the Indian Ocean Dipole (IOD), and found that the IOD-intervened temperatures are highly correlated with Australian temperatures (correlation = 0.87). This is a well-known effect (e.g. Cai et al., 2009), and is accurately modelled by PICABU.
>
> In further response to the questions raised by the reviewer, we unfortunately cannot answer such questions as “Does manually setting the latent variable to a new, higher value represent a 10% reduction in trade winds or an increase in subsurface warm water upwelling?”, since we do not include these variables explicitly in the model. We still believe our approach provides some level of physical insight that we agree could be extended in future work by including more variables.
>
> **We hope that the reviewer’s concerns are addressed by our responses. Thank you very much for this review - your comments have significantly helped us to improve the manuscript and analysis.**

---

> > ### Comment · Reviewer_5AXT · 2025-08-06
> >
> > I would like to thank the authors for their detailed response and for comprehensively supplementing experiments on counterfactual reasoning and the variance of the Bayesian filter. With the additional clarifications on how their method numerically manipulates the latent variables, its viability is more substantiated.
> >
> > My only remaining concern is that incorporating these revisions might result in substantial changes to the original submission. I will forward both my resolved and unresolved concerns to the Area Chair for their consideration. Thank you once again.

---

> > > ### Author Response · Authors · 2025-08-06
> > >
> > > We thank the reviewer again for their valuable feedback.
> > >
> > > We understand that answering the points raised by the reviewer led to substantial updates to the manuscript, and helped improve its quality significantly. However, we think it is overall an appropriate amount of changes:
> > >  - The justification for using the single-parent assumption will result in an additional paragraph.
> > >  - Section 4.4 (counterfactual experiments) will be detailed further, and complemented by additional experiments in the appendix.
> > >  - We will add a paragraph and a table to illustrate the results when using the estimated variance in the Bayesian filter
> > >  - The clarification on hyperparameter tuning will result in an additional paragraph.
> > >
> > > As the single-parent assumption and additional counterfactual experiments are the main concerns shared by other reviewers as well, this will overall be a reasonable amount of changes, and the main structure, takeaways and results presented in the original submission will not change.
> > >
> > > We hope that this resolves the final concern on substantial changes to the original submission and thank the reviewer again for strengthening our work through very valuable feedback. We welcome any further questions and are happy to engage in continued discussion.

---

### Note · Authors · 2025-08-12

We would like to thank the reviewers once again for their thoughtful comments and for engaging with our rebuttals. We are pleased that the reviewers noted several valuable contributions of our work, including the “exceptional novelty” of our Bayesian filter, the strong interpretability of the model, and the subsequent capacity for robust counterfactual experiments. We are also happy that the reviewers appreciated our comprehensive ablation studies and the analysis on synthetic as well as real world data.

The responses suggest that all concerns and questions from the initial reviews have been addressed. In particular, we believe that the additional counterfactual experiments have further strengthened the case that our tool is useful for climate scientists (as discussed by reviewers yBUD and fpGY), as these results show that interventions in observation space directly correspond to physical knowledge about teleconnections.

The only potential remaining concern that we see following the rebuttal period, raised by reviewer 5AXT, is that incorporating the rebuttal experiments may lead to substantial changes from the original submission. However, we do not believe the additional experiments will lead to major changes to the manuscript. The new counterfactual experiments will be described briefly in Section 4.4 (about half a page), with additional details in the Appendix, including the relationship between interventions in latent and observation space. We do not think this changes the original manuscript substantially, as the takeaways and results shown in the main text will not change, but will be better explained and justified thanks to the reviewers’ suggestions. The other points discussed during the rebuttal phase will improve on the existing text. We do not believe these modifications substantially change the paper from the original submission, beyond what would typically be expected from a productive rebuttal discussion.

Again, we are very grateful to the reviewers for engaging with this work and for their thoughtful reviews. It helped us better clarify our exposition and strengthen our analysis.

---

### Decision · Program_Chairs · 2025-09-17

**Decision:**

Accept (poster)

**Comment:**

The paper proposes a Bayesian filtering framework for causal climate emulation, offering a principled approach to capturing dynamical dependencies in climate processes. The idea is novel, timely, and well-grounded, with strong empirical validation across relevant datasets. Reviewers appreciated the clarity of presentation and the methodological contribution, with remaining concerns being relatively minor and not undermining the core contribution. The need to incorporate additional experiments conducted during the rebuttal is not, by itself, a sufficient reason for rejection. Overall, the paper makes a valuable and well-supported addition to the literature, and I recommend acceptance.